# Hydrodynamic assembly of two-dimensional layered double hydroxide nanostructures

Nicholas A. Jose[1,2], Hua Chun Zeng [2,3] & Alexei A. Lapkin[1,2]

Formation mechanisms of two-dimensional nanostructures in wet syntheses are poorly understood. Even more enigmatic is the influence of hydrodynamic forces. Here we use liquid flow cell transmission electron microscopy to show that layered double hydroxide, as a model material, may form via the oriented attachment of hexagonal nanoparticles; under hydrodynamic shear, oriented attachment is accelerated. To hydrodynamically manipulate the kinetics of particle growth and oriented attachment, we develop a microreactor with high and tunable shear rates, enabling control over particle size, crystallinity and aspect ratio. This work offers new insights in the formation of two-dimensional materials, provides a scalable yet precise synthesis method, and proposes new avenues for the rational engineering and scalable production of highly anisotropic nanostructures.

[1] Department of Chemical Engineering and Biotechnology, University of Cambridge, Cambridge Philippa Fawcett Drive CB3 0AS, UK. [2] Cambridge Centre for Advanced Research and Education in Singapore Ltd., 1 Create Way, CREATE Tower #05-05, Singapore 138602, Singapore. [3] Department of Chemical and Biomolecular Engineering, Faculty of Engineering, National University of Singapore, 10 Kent Ridge Crescent, Singapore 119260, Singapore. Correspondence and requests for materials should be addressed to A.A.L. (email: aal35@cam.ac.uk)

Two-dimensional (2D) structured materials, such as graphene, metal chalcogenides, and layered double hydroxides (LDHs), have been suggested as next-generation materials in catalysis[1], energy storage[2,3], and electronics[4]. While some 2D materials are made via vapor-phase deposition, many scalable and low-cost syntheses take place in the liquid phase. The current state of fundamental understanding of how highly anisotropic nanoparticles crystallize and how synthesis conditions affect crystal morphology, size, and structure is rather limited, yet such knowledge is crucial to our ability to rationally design new materials and processes for large-scale manufacturing and applications.

A key problem in identifying the crystallization mechanism of nanostructures is that classical models are overly simplistic, assuming constant surface and bulk energies, while disregarding particle interactions, mass transfer effects, competing kinetics, and external forces. Non-classical mechanisms, including oriented attachment (OA) and two-step nucleation pathways, may dominate when molecular and interparticle interactions are anisotropic. These are important not only in materials synthesis but also in biogenic and geologic environments, for example, in the mineralization of complex calcium carbonate structures in aquatic organisms[5].

In wet syntheses, hydrodynamics govern mixing, residence time, and particle interactions but are often neglected in laboratory studies. Reactor geometry and agitation influence the rates of mass transfer, nucleation and growth, residence time distribution, and the resulting particle size distribution[6,7]. Hydrodynamic shear stress, which is introduced by agitation or pumping, can decrease kinetic interaction barriers to accelerate nucleation and aggregation[8,9]. Shear also induces the alignment of anisotropic particles by stabilizing angular orientations in the direction of flow[10,11]. Conversely, by introducing strain, shear may also inhibit crystallization and causes breakup[12,13]. Depending on its intensity, shear may be used to exfoliate 2D materials like LDH[14] and graphene[15] or induce structuring[16]. The effects of shear on non-classical mechanisms of growth such as self-assembly have yet to be fully elucidated, though recent simulations have shown that the OA of patchy trimer particles is accelerated under certain shear rates[17].

LDHs are cationic 2D metal hydroxides intercalated with anions and possess applications in catalysis[18], drug delivery[19], and energy storage[20]. Colloidal LDH is synthesized via hydrothermal treatment or co-precipitation, usually followed by thermal aging to improve crystallinity[21]. The crystallization process is not well understood and is thought to proceed via local dissolution and crystallization of amorphous LDH[22,23] or via the physical rearrangement of nanocrystalline domains[24,25]. Surfactant-assisted methods[26] and continuous flow reactors have recently been used to generate monolayer LDH nanoplatelets[27–29], though the control of particle size, crystallinity, and thickness remains difficult.

Hydrodynamics also play a key role in the synthesis of LDH. Reactor geometries (e.g., the T-micromixer[30], flow hydrothermal reactor[31], and inline dispersion precipitator[27]), flowrates, and mixing speed have pronounced effects on the crystallite size, surface area, and aspect ratio. However, the physical mechanism behind these effects has not been rigorously investigated. The obscure crystallization mechanism, observed sensitivity to hydrodynamics, and relative ease of synthesis via co-precipitation make LDH a perfect model system for the study of 2D structures under controlled shear.

In this study, we seek to identify the crystallization mechanism of 2D LDHs and examine the effect of hydrodynamics, particularly shear rate. For this, we develop a scalable microreactor, in which a high shear microfluidic environment with microsecond mixing is generated using two-phase annular flow. We observe particle dynamics using liquid transmission electron microscopy (LTEM), a powerful tool for characterizing nanoparticle crystallization and self-assembly[32–34] and reveal that 2D LDH nanoparticles may crystallize via OA. We mechanistically explain the effects of shear on nucleation, growth, and oriented assembly using insights gleaned from LTEM, atomic force microscopy (AFM), powder X-ray diffraction (XRD) and shear-induced aggregation kinetic modeling. At high shear rates, crystallization of 2D nanoplatelets via OA may be accelerated, affecting the resulting particle size distribution and shape.

## Results

**Microreactor fluid dynamics.** The reactor consists of three quartz tubes placed in a staggered, coaxial configuration; see Fig. 1a and Methods. A high velocity gas is pumped through the innermost tube, with liquid reagent streaming through the two outer tubes, creating two regions of two-phase annular flow. The mixing region forms by the collision and hydrodynamic focusing of two-liquid annular flows, which then stabilize into a wavy annular thin film. As the thin film thickness stabilizes, the shear rate $\dot{\gamma}$ in an annular thin film approaches a constant value, which may be determined from the film thickness and velocity or gas and liquid flowrates[35].

The liquid profile, mixing dynamics, and mixing time were measured with high-speed fluorescence confocal microscopy. As seen in Fig. 1b, the annular film thickness decreases from 100 to 10 μm with increasing $\dot{\gamma}$, indicating a highly confined reaction environment at shear rates up to $10^6\,\mathrm{s}^{-1}$. These measurements agree with modeled predictions[35]. Further discussion of the velocity flow field and decomposition of the gradient and rate-of-strain tensors is included in Supplementary Discussion – Gradient Tensor Decomposition.

When Liquid 1 stream is laced with a fluorescent tracer, as seen in Fig. 1c, micro-vortices are seen originating at the tip of the first annular flow outlet, likely generated by Kelvin–Helmholtz instabilities at the high shear interface, and by secondary flows as the flow suddenly accelerates in the radial direction. These vortices grow and dissipate in a mixing process known as vortex engulfment, a well-described mode of micromixing[36,37]. After the transient mixing section, the liquid profile stabilizes into wavy striations. High-speed video footage is available in Supplementary Movie 1.

The measured characteristic micromixing times, which range from 11 to 0.4 ms, are in close agreement with those predicted by an engulfment mixing model[36], as seen in Fig. 1d, confirming that the engulfment of microscale eddies is the dominant mixing mechanism. The energetic efficiency of mixing is on average 53% of the theoretical efficiency, an improvement over rotor-stator mixers (~35%) and serpentine micromixers (~14%)[38], which could reduce power consumption at the industrial scale. A comparison with other mixing technologies is given in Supplementary Table 1, where the annular microreactor is one of the fastest mixing techniques for the synthesis of nanoparticles. It is slower than highly confined single-phase microreactors such as the turbulent tangential mixer and capillary ball mixer, though it has significantly less clogging risk and a lower pressure drop.

Macromixing times, which were determined from the concentration of fluorescent tracer, decrease from milliseconds to microseconds as $\dot{\gamma}$ increases; the fastest mixing time measured was 288 μs, as seen in Fig. 1d. These are in the same range as micromixing times determined by Villermaux–Dushman chemical test method[39,40] and in agreement with predictions from the engulfment mixing model. Rapid micromixing and macromixing timescales ensure that parallel reaction systems with different

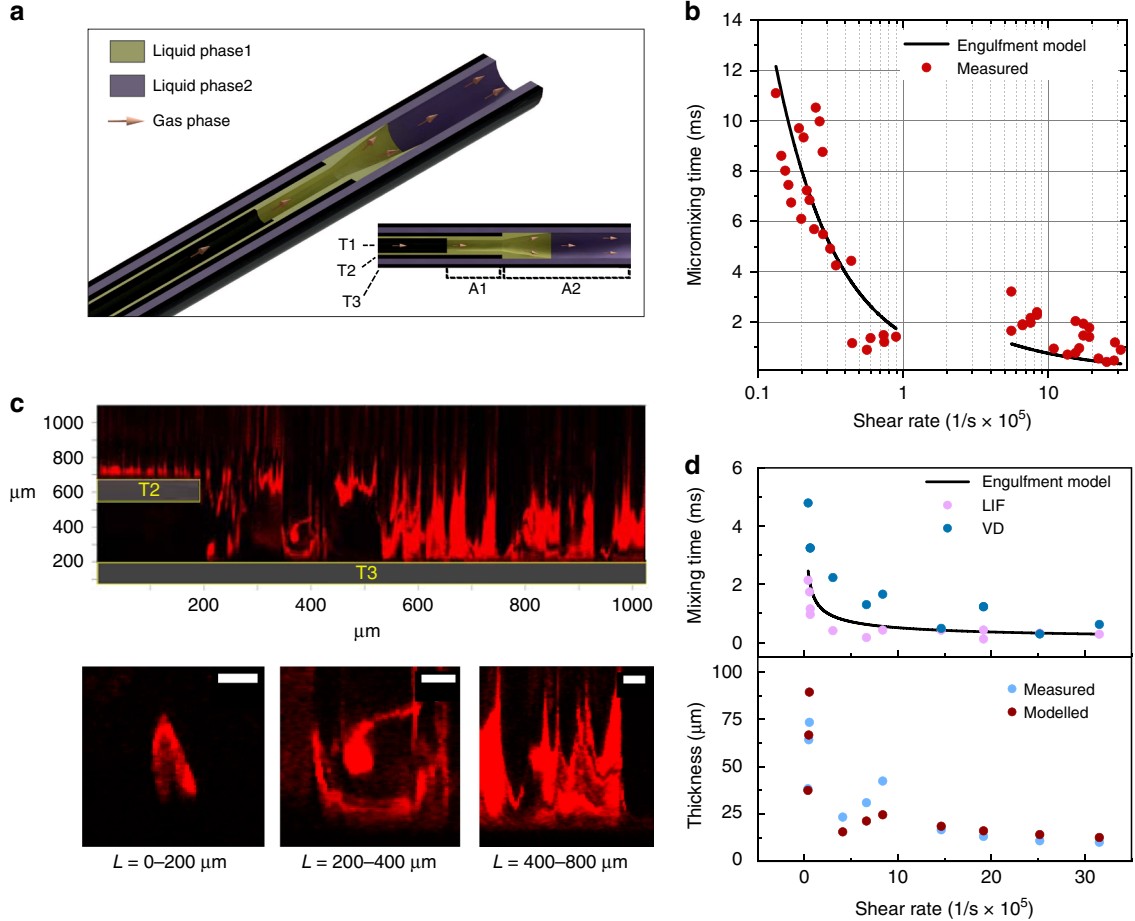

**Fig. 1** Annular reactor mixing dynamics. **a** Schematic of microreactor axial cross-section, where T1, T2, and T3 are the three coaxial tubes, and regions A1 and A2 are the annular flow regions created by the flow of the gas phase (red arrow). In A2, mixing of phase 1 (yellow) and phase 2 (purple) occur. **b** Predicted and measured characteristic micromixing times using Villermaux–Dushman test reaction. **c** High-speed laser induced fluorescence (LIF) microscopic frames of mixing in the A2 region. Phase 1 (red, laced with fluorescent tracer) and Phase 2 (blank) flow from left to right. At top is half of the axial cross-section, where T2 and T3 are highlighted, showing chaotic mixing over 1 mm. Below are snapshots of vortices along the reactor length, which are initially generated at the edge of T2 (scale bar = 10 μm). **d** Measurements and model predictions of macromixing time and film thickness (bottom) in A2 as a function of shear rate. The mixing time calculated from LIF microscopy measurements (LIF) is compared to the mixing times from Villermaux–Dushman test reaction (VD) and predictions from the engulfment mixing model

rates, such as nucleation (fast) and growth (slow), may be segregated and that the residence time distribution should be narrow, ensuring a product with small size and low polydispersity.

There are several key advantages that make this reactor amenable for rapid scale-up and precise manufacturing of nanostructures: (1) with microsecond mixing, fast kinetics such as nucleation, surface growth, and aggregation during high supersaturation precipitation may be controlled; (2) with high wall shear and only one non-slip boundary, scaling and clogging are reduced; (3) increasing throughput can be achieved by increasing the number of reactors or the flowrate, in accordance with previously determined reaction conditions; and (4) as a continuous reactor, space–time yield is high, which reduces capital and operating costs during scale-up[41,42], integration with downstream processes is simplified, and inline characterization/optimization is possible[43,44]. This flexibility provides a means of precisely controlling hydrodynamic shear, mixing time, and residence time from the laboratory bench to the industrial scale and may be used to intensify the development of colloid synthesis techniques of nanomaterials.

**LDH synthesis and characterization.** Colloidal dispersions of MgAl LDH (hydrotalcite) were co-precipitated from an aqueous solution of metal nitrates and a basic solution of sodium hydroxide and sodium carbonate at a supersaturation ratio of $3 \times 10^4$ at 21 °C. Reagents and air were injected at flowrates corresponding to shear rates ranging from $7.26 \times 10^4$ to $3.16 \times 10^6 \, s^{-1}$. The suspensions consisted of fine, white, translucent precipitate, which formed a thick gel after centrifugation. After rinsing with deionized (DI) water, the colloidal stability of the material was increased, as has been observed in previous studies[27].

AFM measurements were conducted to determine the particle size distribution in dry samples, measuring both height ($H$) and area-averaged diameter ($D$). AFM height maps showed nanoplatelets of $28.1 \pm 5.5$ nm in diameter, ranging from 0.7 to 1.5 nm in height, corresponding to monolayer and bilayer LDHs. These nanoplatelets would also form clusters, seen in Fig. 2a. The aspect ratios (AR = $D/H$), measured for each particle and cluster strongly correlate with their height. The slope of this correlation varied non-monotonically with $\dot{\gamma}$; at $8.4 \times 10^5 \, s^{-1}$ "flat" clusters of maximum anisotropy formed, while at lower or higher shear rates "round" clusters of lower anisotropy formed, which is shown in Fig. 2b.

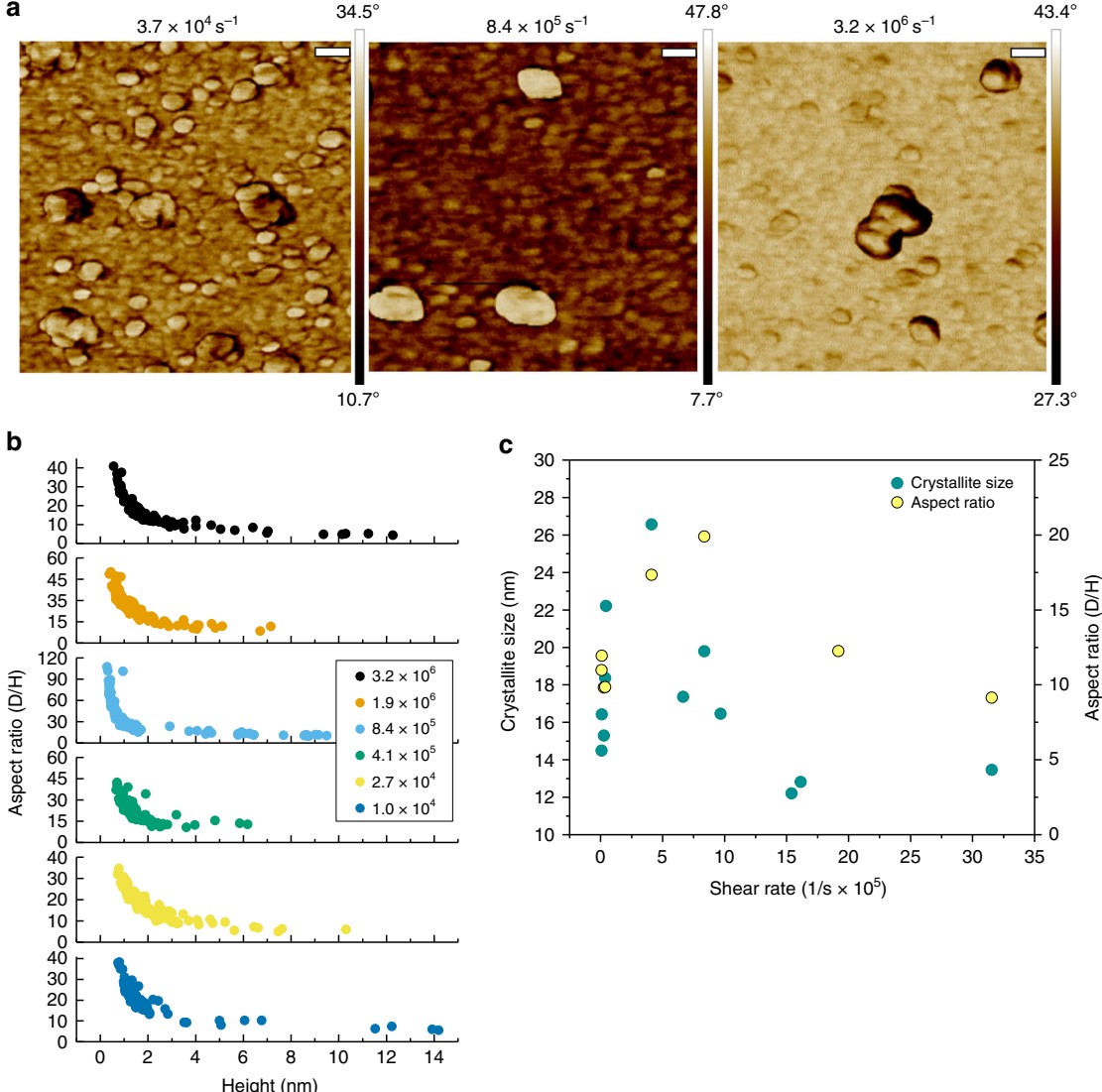

**Fig. 2** LDH size, shape, and crystallinity at different shear rates. **a** AFM phase maps of nanoplatelets and aggregates synthesized at increasing shear rates, showing variations in shape (scale bar = 50 nm). **b** Trends of individual particle aspect ratios (AR) plotted against height (*H*) and shear rate. **c** Aspect ratio (at a thickness of 3.85 nm) and crystallite size plotted shear rate

The correlation between aspect ratio and height indicates that the particles assemble in an ordered manner, which directly verifies previous studies that have suggested that LDH particles may preferentially orient epitaxially, edge-to-edge[45] and face-to-face[24]. More interestingly, the non-monotonic dependence of aspect ratio and shear suggests that shear not only influences the kinetics of OA but also has competing effects that enhance or decrease particle anisotropy.

XRD and high-resolution TEM (HRTEM) were performed to identify the scale at which shear governs cluster anisotropy. XRD patterns showed the characteristic peaks of hydrotalcite structure at $2\theta = 11.5°$, $22.9°$, and $61.8°$, corresponding to the (003), (006), and (110) lattice planes, as seen in Supplementary Figure 1. The crystallite size in the [110] direction (the basal crystal plane) as determined using the Scherrer equation, varied non-monotonically with $\dot{\gamma}$, similarly to the trend in aspect ratio observed with AFM, as shown in Fig. 2c. From $7.26 \times 10^4$ to $4.11 \times 10^5 \, s^{-1}$, the crystallite size increased from 15 to 27 nm, approximately the observed diameter of the primary platelets, and decreased to 13 nm at $3.16 \times 10^6 \, s^{-1}$. This suggests that shear affects particle crystallinity, which then governs their oriented

aggregation and anisotropy. Highly crystalline particles synthesized at $4.11 \times 10^5$–$8.4 \times 10^5 \, s^{-1}$ align readily, whereas less crystalline particles do not.

HRTEM images, exhibited in Fig. 3a, b, showed that LDH synthesized at $4.11 \times 10^5 \, s^{-1}$ were indeed highly crystalline. The LDH synthesized at $3.16 \times 10^6 \, s^{-1}$ was polycrystalline, with domain sizes in the order of 5 nm, seen in Fig. 3c. The meso-crystalline structure of the densely packed aggregates was also resolved. Displayed in Fig. 3d are layers of LDH primary particles separated by 0.8 nm, indicative of the (003) spacing. A fast-Fourier transform shows visible (003) and (106) peaks, revealing that not only are the particles stacked but are also epitaxially attached in the basal plane.

**Liquid transmission electron microscopy.** LTEM was performed to visualize the mechanism of platelet aggregation and the mechanism by which hydrodynamic shear may affect crystallization kinetics. Immediately after synthesis, suspensions were diluted, loaded into a 150-nm-thick silicon nitride cell using a Protochips liquid holder, and imaged within an hour (see

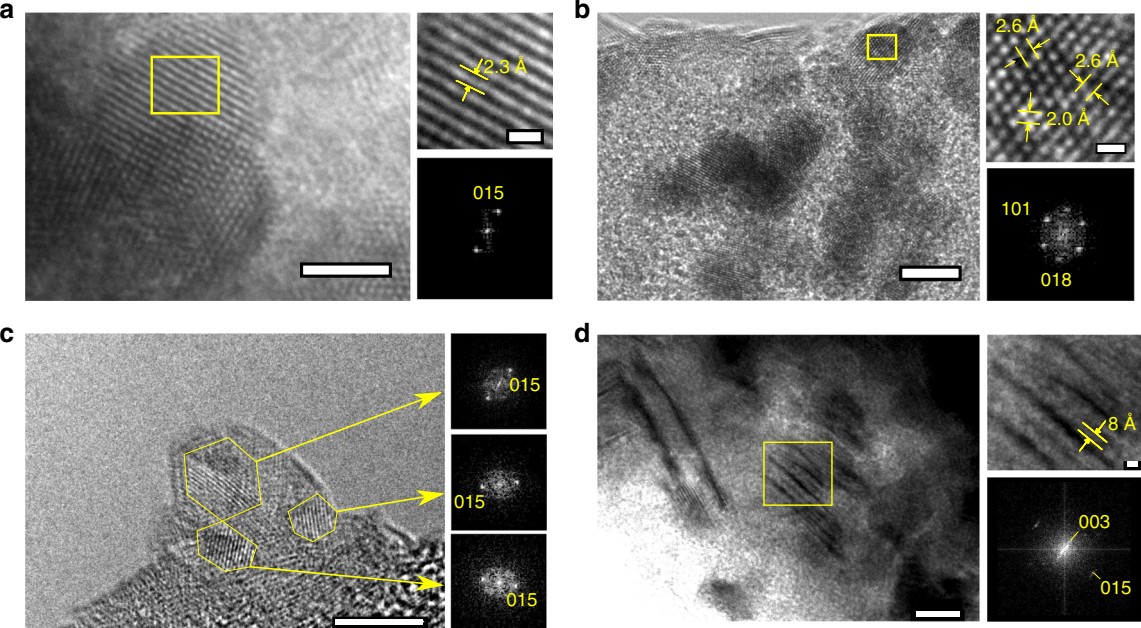

**Fig. 3** HRTEM images of nanoplatelets. **a** Particles approximately 5 nm in diameter with high crystallinity on a SiO substrate, synthesized at $4.11 \times 10^5$ s$^{-1}$ (left). Highlighted crystalline domain (upper right) and corresponding Fourier transform (lower right) show lattice spacing of 0.23 nm, indicative of the (015) reflection. Scale bars are 2.5 nm (left) and 0.5 nm (upper right). **b** Uniform and crystalline particles synthesized at $4.11 \times 10^5$ s$^{-1}$, approximately 5 nm in diameter aligned on a SiO substrate (left). Highlighted crystalline domain (upper right) and corresponding Fourier transform (lower right) show lattice spacings of 0.26 and 0.2 nm, indicative of the (101) and (018) reflections, respectively. Scale bars are 5 nm (left) and 0.5 nm (upper right). **c** A nanoplatelet 7 nm in diameter synthesized at $3.16 \times 10^6$ s$^{-1}$ (left). Fourier transforms of the outlined crystal domains (upper, middle, and lower right) indicate variation in the position of the (015) peak and crystallographic orientation. Scale bar is 5 nm (left). **d** Mesocrystalline structure in a large aggregate (left), with highlighted crystalline domain (upper right). Scale bars are 10 nm (left) and 1 nm (upper right). Interlayer stacking between particles is approximately 0.8 nm, indicative of the (003) plane. Corresponding Fourier transform (lower right) also shows a peak indicative of the (015) plane

Methods). Dispersed hexagonal disks ≤10 nm in diameter adhered to the silicon nitride membrane surface. These were predominantly oriented face down, most likely due to Coulombic attraction between the positively charged hydrotalcite and negatively charged silicon nitride, limiting Brownian motion. We speculate that these LDH particles most likely aggregate to form the ~30 nm nanoplatelets measured in AFM during the post-processing steps of centrifugation, rinsing, and drying.

Under static conditions, particles at close distances aggregated by "walking" to each other, rotating until crystal facets were aligned, and attaching edge-to-edge, as seen in Fig. 4a and Supplementary Movie 2. As seen in Fig. 4b, the speed of approach was correlated to the alignment of facets, clearly indicating a mechanism of OA. In one interesting scenario seen in Supplementary Movie 3 and Supplementary Figure 2, six particles with diameters of approximately 5 nm are loosely packed in a pentagonal configuration. These six particles undergo much slower rates of aggregation, despite their proximity. While the high local concentration of particles can decrease Brownian diffusivity, OA is also likely hindered by the pentagonal configuration of these particles, which limits facet alignment with the neighboring particles. This also indicates that misalignment, in addition to electrostatic repulsion, is a significant barrier to aggregation where van der Waals forces are the main driving force.

The introduction of flow accelerated particle aggregation. Local particle motion was difficult to induce with an external syringe pump; however, when a bubble was created via radiolysis under an intensified electron beam[46], the de-wetting of the silicon nitride membrane created a local flow strong enough to convect particles. The approach rates of two colliding particles, which was determined by tracking interparticle distances, were compared for static and bubble-induced flow conditions. The approach velocity in aggregation events between two ~8 nm particles in a static liquid environment and in a flowing liquid are detailed in Fig. 5 and Supplementary Movies 4 and 5. The average approach velocity was about four times faster in a flowing liquid as it was in a static liquid at interparticle distances under 25 nm.

This acceleration may be explained by two factors: flow-induced convection and rotation. Flow increases mass transport via advection and the dissipation of energy into heat, which enables particles to overcome the electrostatic barrier to aggregation. This can be modeled as a lowered effective kinetic barrier[12,47–49]. In static conditions, attraction to the TEM cell membrane surface limits rotation. When hydrodynamic forces overcome this adhesive force, particles rotate more easily and as a result will orient more quickly. This effect seems to be important at very close distances, at the order of the particle size. The added barrier to aggregation from misalignment may perhaps be included in models as a modified collision frequency parameter.

These are, to the best of our knowledge, the first direct observations of LDH crystallization via OA of nanoplatelets and of hydrodynamically accelerated OA via bubble-induced flow in LTEM. We speculate that this is the dominant mechanism in concentrated, rapidly mixed conditions and may be a significant crystallization pathway during aging. This confirms the previous studies that have suggested an OA mechanism[24,29] in addition to Ostwald ripening, a slow dissolution, and recrystallization process[22,50].

The anisotropy of LDH's surface is most likely responsible for its propensity to undergo OA. While the hydroxide groups at the basal surface of LDH sheets are triply coordinated, the edge

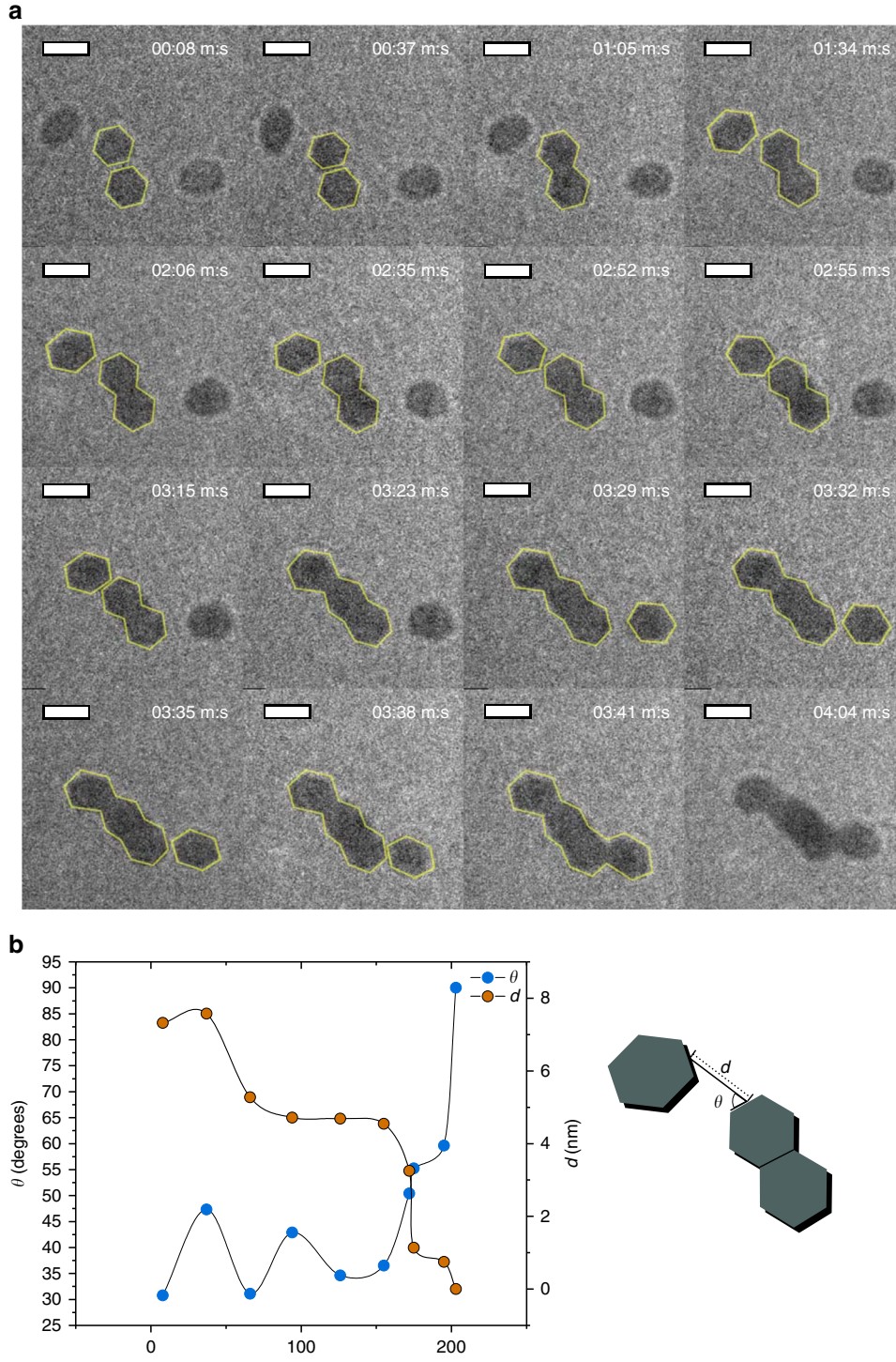

**Fig. 4** Oriented attachment of LDH nanoparticles, observed via LTEM. **a** LTEM frames showing the oriented aggregation of hexagonal particles (outlined). Scale bar is 10 nm. **b** Left, the angular orientation $\theta$ and distance $d$ between facets of attachment for the aggregation event observed from 1:34 to 3:23 (m:s), where lines are added to guide the eye. Right, a schematic showing measurements of $\theta$ and $d$

hydroxide groups are bi- and mono-coordinated, which are labile and amphoteric[51,52]. These groups are responsible for the chemisorption of heavy metal ions, such as arsenate and chromate. In suspensions with high pH values (10–11), such as the synthesis conditions used in this study, these edges decrease in charge via dissociation of hydroxyl groups and adsorbed water, decreasing Coulombic repulsion such that aggregation at the edges is kinetically favored.

Adsorbed anionic species may also play a significant role in OA by mediating the acid/base interactions of edge hydroxyl groups. For example, amphoteric polyprotic anions such as carbonate increase the aggregation and crystallization of LDH[53,54], presumably by both screening the surface charge of LDH while retaining its amphoteric nature. Conversely, using ligands and/or solvents that inhibit acid/base interactions may inhibit OA and increase the basal/edge interactions. This could be the primary

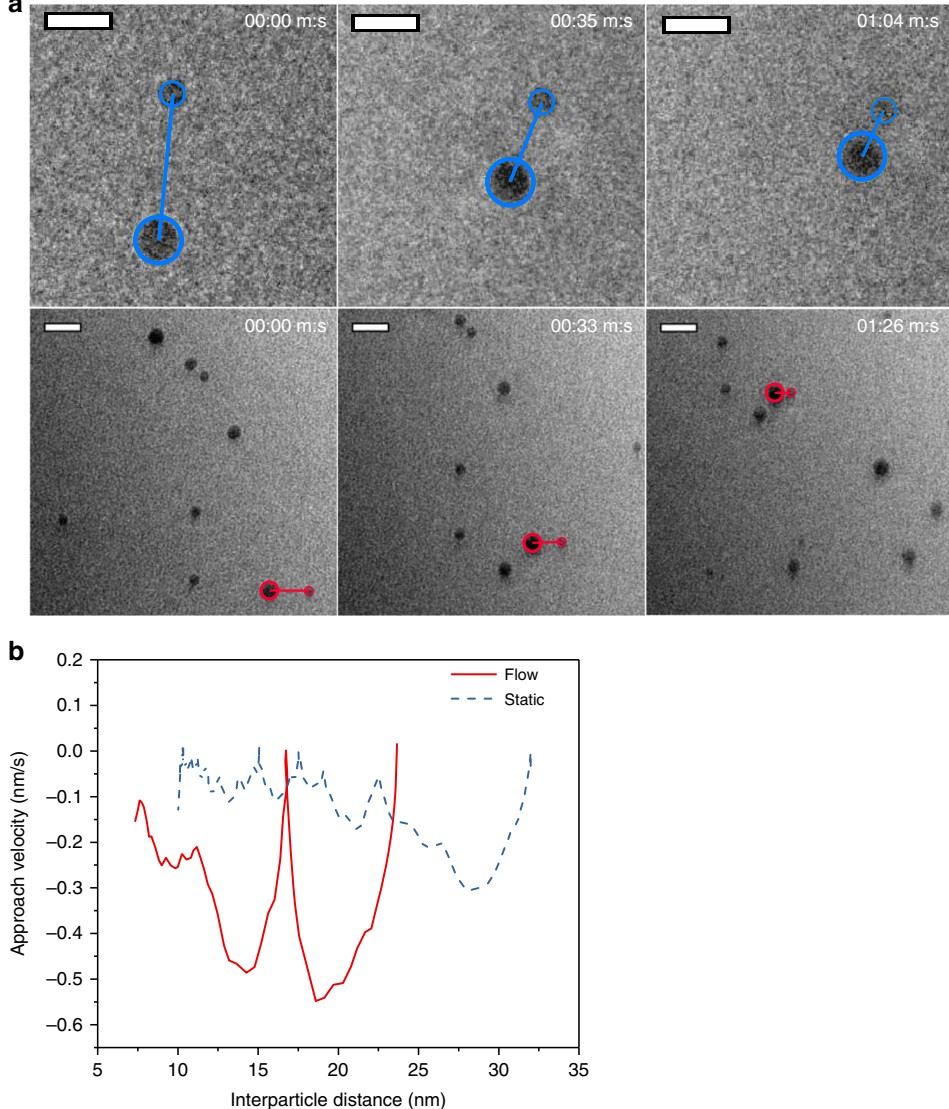

**Fig. 5** Attachment of particles in static and flowing conditions. **a** At top are particles aggregating in static liquid (scale bar = 10 nm) and at bottom are particles aggregating in bubble-driven flow (scale bar = 20 nm). Particles of interest and their interparticle distance are outlined. **b** The approach velocity plotted against interparticle distance in flow and static fluids, indicating accelerated aggregation in flow

mechanism behind the generation of flower-like morphologies when LDH is synthesized using anions like organic acids[55] and organic solvents like acetone[56]. Further exploring the effects of anionic species and solvent interactions on OA in LDH would be useful in the rational assembly of LDH nanostructures.

The observation of dynamic particle movements, like walking and rotation, also introduces the use of LDH particles to quantitatively study these non-DLVO factors. It may also be possible to explore the formation mechanisms of dense or fractal aggregates, which were also observed (see Supplementary Figures 3 and 4 and Supplementary Discussion-Aggregate Formation). In such studies, it would be important to carefully consider the effects of the electron beam on the local ionic environment, particle charge, and silicon nitride surface.

**Kinetic analysis**. The particle size distributions of LDH synthesized at $4.1 \times 10^4 \, s^{-1}$ (A), $8.4 \times 10^5 \, s^{-1}$ (B) and $3.2 \times 10^6 \, s^{-1}$ (C) were measured to determine the effect of shear on nucleation and growth kinetics. The particle areas and calculated hexagonal side length $a$ are compiled in Fig. 6a. From A to B, the median particle

area decreases from 43.4 to 39.0 $nm^2$ and the side length distribution narrows from 3.92 ± 1.03 to 3.89 ± 0.94 nm. From B to C, the area distribution becomes multimodal and the side length distribution widens to 3.75 ± 0.97 nm. By deconvoluting the area distribution of C, we find five peaks regularly separated by 11.7 $nm^2$, approximately the area of the smallest particle measured (10.6 $nm^2$). This particle, as seen in Fig. 6b, possesses a side length of 2.14 nm and is composed of only 113 cations.

We speculate that this is the critical nucleus size, above which particles grow via surface addition or aggregation. This explains why LDH nanoparticles as small as 4 and 7.8 nm have been observed in previous studies[27,29,57]. This critical size is influenced by a range of factors such as the initial supersaturation, temperature, and pH, which are explained by classical nucleation theory. The size may also represent a non-classical, or metastable, configuration at which the surface is sufficiently stabilized by complex local solvent–solute–surface interactions[5]. This seems likely, as aggregates anneal slowly after attachment, which is seen Supplementary Figure 4, even under the excitation of the electron beam. The structure and evolution of these interactions are still not well known and are a subject of future study.

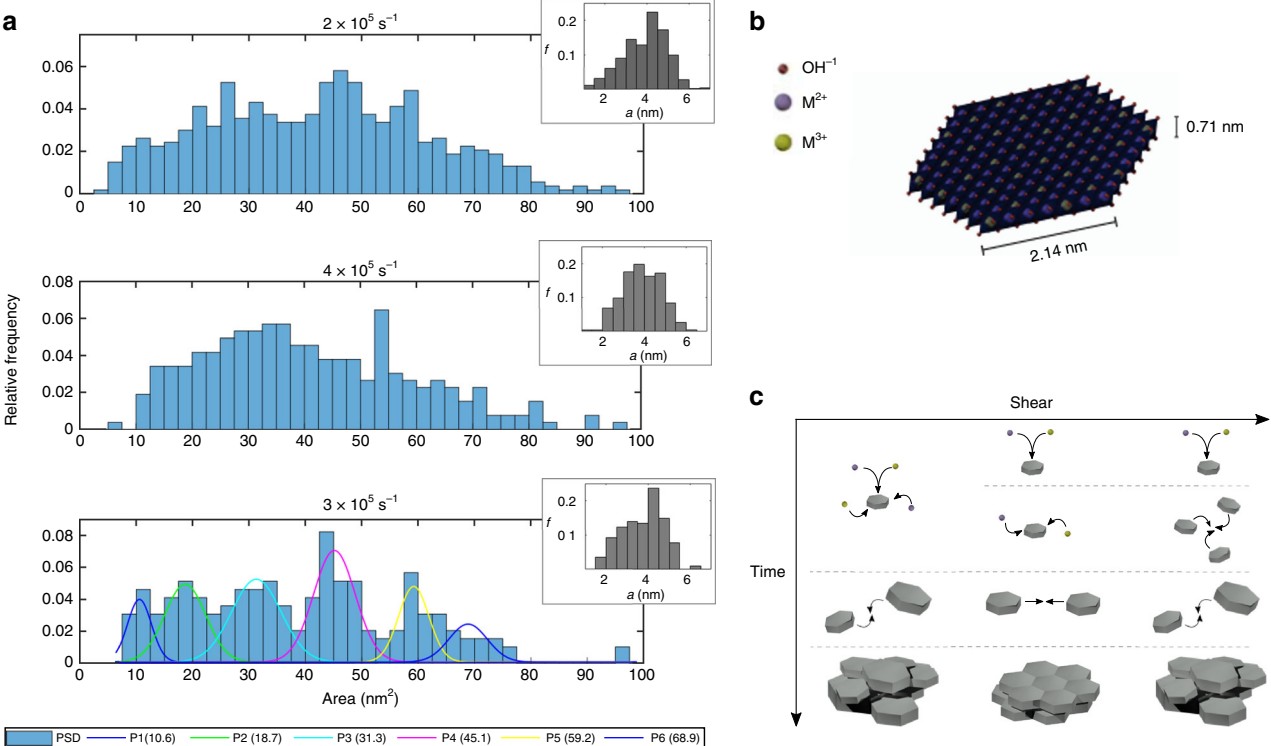

**Fig. 6** Particle size and kinetic analysis. **a** Relative frequency $f$ distributions of particle areas and calculated side lengths $a$ measured in LTEM. Deconvoluted peaks are shown in the area distribution at $3 \times 10^6$ s$^{-1}$. **b** Crystal structure of a monolayer primary particle with a side length of 2.14 nm, determined from the spacing of peaks at $3 \times 10^6$ s$^{-1}$. **c** Schematic of crystallization processes under increasing shear rate. At low shear rates (left), nucleation and surface growth occur simultaneously, after which polydisperse primary particles aggregate in a disordered fashion, leading to polycrystalline aggregates. At higher shear rates (middle), enhanced mixing leads to the segregation of nucleation and surface growth, after which monodisperse particles align into ordered crystalline aggregates. Above a critical shear rate (right), oriented aggregation is the dominant growth process, leading to a polydisperse population of primary particles that form polycrystalline aggregates

We interpret these changes in particle size distribution by considering the effects of shear on mixing, kinetics, and thermodynamics. For this analysis, we use the concept of characteristic aggregation timescales based on previously developed kinetic rate theories for the growth and aggregation of Brownian particles, which are described in Methods. At $4.1 \times 10^4$ s$^{-1}$, the mixing time is approximately 4.4–2.1 ms, which is lower, but on the same order as the characteristic growth time 16.6 ms. Therefore, nucleation and surface growth may occur concurrently, producing a more polydisperse population. The characteristic aggregation time for two nuclei, which we approximate as Brownian discs 0.7 nm in thickness and 2 nm in radius is 0.68 s, implying that aggregation is not a dominant growth mechanism.

At $8.4 \times 10^5$ s$^{-1}$, the mixing time decreases to 1.9–0.43 ms, which is faster than the timescale of surface growth, resulting in the effective segregation of nucleation and growth and a more homogeneous size distribution. The calculated characteristic aggregation time is 150 ms, which is an order of magnitude greater than the growth timescale.

At $3.2 \times 10^6$ s$^{-1}$, the mixing time is <1 ms, ensuring segregation of nucleation and growth steps, and the shear rate is significant enough to affect the motion of colloidal particles <10 nm. The Peclet number (Pe) represents the ratio of hydrodynamic forces to Brownian forces, where at Pe<<1 hydrodynamic forces may be neglected. The rotational Peclet number (Pe$_r$) reflects the effect of hydrodynamic forces on particle rotation, where at Pe$_r$<<1 particle rotation is random; at higher Pe$_r$ particles may align in the direction of flow (see Methods). For disc-like particles of 4 and 8 nm in diameter, Pe equals 0.07

and 0.46, respectively, and the rotational Peclet number (Pe$_r$) equals 0.06 and 0.47. This implies that hydrodynamic forces may significantly affect the motion and alignment of particles at this shear rate. The calculated characteristic aggregation time for a nuclei is 77 ms, which is at the same order of the growth timescale. Thus aggregative growth via OA is accelerated and occurs simultaneously with surface growth, resulting in a multimodal particle size distribution. This aggregation time may be even lower due to flow-induced alignment; this effect could be estimated by developing a generalized kinetic model for the OA of highly anisotropic nanoparticles in shear flow, which is outside the scope of this study. Further discussion of how the velocity field affects particles via gradient tensor decomposition and an alternate approach for calculating particle growth rate using the Sherwood number are included in Supplementary Information.

Post-synthesis processing steps of centrifugation and rinsing most likely caused these particles to aggregate into the stable 30 nm platelets seen in AFM. Because the population of particles formed at $8.4 \times 10^5$ s$^{-1}$ was more homogeneous, it readily formed ordered assemblies, which crystallized to form the highly anisotropic particles. More polydisperse populations, formed in poorly mixed conditions at $4.1 \times 10^4$ s$^{-1}$ or in aggregation-dominated conditions at $3.2 \times 10^6$ s$^{-1}$, formed more disordered assemblies, creating polycrystalline particles. These polycrystalline particles then clustered in a more disordered fashion, with lower anisotropy. This behavior is attributed to the increase in surface free energy required to form ordered assemblies, slowing the kinetics of crystallization[58,59]. A schematic of this process is illustrated in Fig. 6c.

## Methods

**Annular microreactor construction**. Three quartz tubes were installed in a tube-in-tube configuration to create two annular flow zones (see Fig. 1a). In section A1, air flow through T1 forced Liquid Phase 1 into a thin film at the inner walls of T2. In A2, the thin film of Liquid 1 from A1 is then contacted with a thin film of Liquid Phase 2. In T3, mixing and reaction occur. The smooth surfaces of the quartz tubes increase flow stability and decrease scaling. They are also amenable to in situ analytical techniques, such as laser scattering, spectroscopy, optical microscopy, and X-ray scattering.

Liquids were delivered to the reactor using a KDS Legato Dual Syringe Pump using disposable plastic 50 or 10 mL Terumo syringes. Compressed dried air was passed through a 200 μm filter (Swagelok) and delivered to the reactor using a Sierra SmartTrak C50L Mass Flow controller (20 L min$^{-1}$ max, 2% accuracy). For low gas flowrates (0.1 mL min$^{-1}$), a laboratory rotameter was used. The pressure of the Liquid 2 stream was monitored inline to determine when flow had reached a steady state.

Stainless steel tee connectors with 1/4" and 1/16" diameter compression fittings were purchased from Swagelok. Precision quartz capillary round tubes of the following dimensions: 0.30 mm inner diameter (ID) × 0.4 mm outer diameter (OD) × 100 mm L (T1), 0.50 mm ID × 0.7 mm OD × 100 mm L (T2), and 1 mm ID × 1.2 mm OD × 100 mm L and 300 mm L (T3) were purchased from VitroCom (supplied by ArteGlass Japan). Gas tight connections between stainless steel fittings and quartz tubes were made with graphite and Teflon 1/16" compression ferrules. Also, 0.8 mm ID and 0.5 mm ID ferrules were purchased from Restek, and 1/16" PTFE ferrules were purchased from Swagelok. PTFE (0.8 mm ID) tubing from ChemiKalie was used for delivery of liquid solutions to the reactor.

**Fluorescence microscopy**. For high-speed microscopic studies, a mount was constructed using a Formlabs 1+3D printer with high temperature resin and a 1 mm quartz glass slide. Channels and tube connections were sealed with rubber silicone sealant and epoxy adhesive. Gas flowrates from 0.4 to 3 L(STP) min$^{-1}$ and liquid flowrates from 2 to 20 mL min$^{-1}$ were evaluated, keeping both inlet liquid streams at the same flowrate.

Confocal fluorescence imaging was performed with a Leica TCS SP5 using a scanning Ar laser at 8000 Hz with ×10 objective magnification. The reactor tube was immersed in glycerol and placed above a 200 μm glass coverslip to reduce the effects of refraction from the tube curvature.

To accurately determine the liquid profile, a dilute fluorescein (Sigma Aldrich) solution was used in both liquid streams. Axial cross-section images were taken over the first 7–8 mm of the reactor at a resolution of 0.57 μm pixel$^{-1}$ and averaged over 64 frames. Owing to the wavy nature of the liquid, the average height was calculated as the distance from the reactor wall to the point at which the intensity was $1/\sqrt{2}$ of the maximum intensity, the root mean square of a sine wave.

To determine mixing dynamics, a dilute fluorescein solution was used in the first annular inlet flow and a blank water stream was used in the second. Video over the first several millimeters of the tube was taken without frame averaging. A median filter with a radius of two pixels was applied to reduce image noise.

To determine macromixing time, a dilute fluorescein solution was used in the first annular inlet flow and a blank water stream was used in the second. The liquid profile was determined as previously described. The average intensity at each tube length was then determined. The mixing length was determined as the point at which the intensity was within a standard deviation of its stable average; this was then divided by the average liquid velocity to determine the effective macromixing time.

**Micromixing characterization**. To experimentally quantify the mixing efficiency of the reactor at various flow conditions, the Villermaux–Dushman method[40,60] was used. This test characterizes micromixing time with a parallel competing reaction system in which the quasi-instantaneous neutralization of boric acid (Eq. 1) by sulfuric acid competes with the reduction of iodate (Eq. 2) to iodine, which forms a triiodide ion (Eq. 3) that can be measured via ultraviolet (UV) spectrometry. In a system with infinitely fast mixing, boric acid will completely neutralize the acid species before iodine may be formed; in a very slowly mixed system, a significant amount of iodine will evolve. By varying reagent concentrations and volumetric fractions, one may tailor the characteristic iodide reaction time to the expected mixing time, with enough sensitivity to detect changes with different flow conditions.

$$H_2BO_3^- + H^+ \leftrightarrow H_3BO_3 \tag{1}$$

$$5I^- + IO_3^- + 6H^+ \leftrightarrow 3I_2 + 3H_2O \tag{2}$$

$$I^- + I_2 \leftrightarrow I_3^- \tag{3}$$

By using the incorporation micromixing model[39], the characteristic mixing time $t_m$ may be determined from the concentration of triiodide. By comparing the measured segregation index to the model, the characteristic micromixing time may be determined with good accuracy in continuous and batch reactors. It should be

noted that, because the known kinetics of this reaction are debated, the characteristic reaction times calculated using this method should only be used to rank mixers characterized using the same protocol[61–63]. In this case, the pH of the mixture was carefully controlled to 7.4 so as to ensure the complete dissociation of sulfuric acid, which has been previously noted as a point of concern[64]. Comparing both the mixing time measured by fluorescence microscopy and mixing time from Villermaux/Dushman test reaction allows for a more universal estimation of the mixing time.

A semi-empirical model of annular flow in microtubes[35] was used to predict the pressure drop, liquid film thickness, shear, and velocity after the transient mixing region. According to the *Re* number, the flowrates used are in the laminar regime. With the predicted pressure drop and velocity, the characteristic mixing time could then be estimated using the relationship between the rate of energy dissipation and micromixing time for high shear flows where vortex engulfment is the dominant mixing mechanism. This relationship is given in Eq. 4[36], which applies for low Schmidt (Sc) numbers «4000 (for water at ambient conditions $Sc = \frac{\nu}{\rho D} = 406$) where $\tau_E$ is the characteristic engulfment time (s), $\nu$ is the kinematic viscosity (m$^2$ s$^{-1}$), $\rho$ is the density (kg m$^{-3}$), $D$ is the diffusion coefficient (m$^2$ s$^{-1}$), in this calculation the self-diffusion coefficient of water[65], and $\varepsilon$ is the rate of energy dissipation (m$^2$ s$^{-3}$).

$$\tau_E = 17\sqrt{\frac{\nu}{\varepsilon}} \tag{4}$$

Potassium iodide, potassium iodate, boric acid, sodium hydroxide, and sulfuric acid were purchased from HCS Scientific. A buffered solution (solution B) of 0.25 M H$_3$BO$_4$, 0.125 M NaOH, 0.005 M KI, and 0.001 M KIO$_3$ and an acidic solution (solution A) of 0.05625 M H$_2$SO$_4$ were prepared in DI water at approximately 21 °C.

Solutions A and B were delivered to the reactor at equal flowrates with compressed dried air to provide total liquid flowrates ($Q_L$) from 0.4 to 20 mL min$^{-1}$ and total gas flowrates ($Q_G$) from 0.1 to 3 L min$^{-1}$. The length of the mixing zone A2 was kept at 50 mm. Product was collected in a vial and then subjected to UV-visible (Uv-Vis) light absorption analysis with an Agilent Cary 60 at 353 nm, diluting if necessary to get values between 0.1 and 0.3. UV-Vis absorption was used to determine the concentration of triiodide (I$_3^-$), which was then used to calculate the concentration of I$_2$. The product pH was approximately 7.65, which is in the optimal zone for this reaction to determine mixing efficiency[66]. Between each experiment, the reactor was flushed with water to avoid cross-contamination.

**LDH synthesis**. Hydrotalcite was synthesized via the co-precipitation method, in which Mg and Al salts are combined with NaOH and Na$_2$CO$_3$[21]. At high super-saturation, precipitation is rapid, ensuring complete conversion within the short timescale used[27].

Magnesium nitrate hexahydrate, aluminum nitrate nonahydrate, sodium hydroxide, and sodium bicarbonate were purchased from HCS Scientific. A metal ion solution (solution A) of Mg$^{2+}$ and Al$^{3+}$ and a basic solution (solution B) of NaOH and Na$_2$CO$_3$ were prepared in DI water at approximately 21 °C. The molar ratio of Mg$^{2+}$: Al$^{3+}$ was fixed at 3:1. A mixture with a supersaturation of $3 \times 10^4$ corresponds to a solution A concentration of 0.112 M Mg$^{2+}$ and 0.0372 M Al$^{3+}$, with a solution B concentration of 0.291 M NaOH and 0.0186 M Na$_2$CO$_3$[23]. Synthesized product suspensions had pH values of approximately 10.65, in which the formation of hydrotalcite is favorable.

While gas flowed through T1, solutions A and B were pumped through T2 and T3, respectively, at equal flowrates. The total liquid flowrate $Q_L$ was varied from 0.4 to 20 mL min$^{-1}$ and the gas flowrate $Q_G$ from 0.1 to 3 L min$^{-1}$. For TEM, AFM, and XRD analysis, the collected suspension was centrifuged immediately after synthesis in a Hanil 514 R Centrifuge at 6000 RPM for 3 min. The collected solids were rinsed three times with DI water to purge unreacted ions. An aliquot was then diluted in ethanol 100× for TEM, XRD, and AFM sampling.

**Transmission electron microscopy**. The ethanol suspensions were ultrasonicated in an Elma Ultrasonic S100h bath for 1 min to disperse large agglomerates, dropped onto Formvar or holey carbon 200 mesh copper TEM grids or SiO-coated copper TEM grids purchased from InLab Supplies, and dried at room temperature. It was found that SiO was a better substrate for stabilizing and dispersing LDH particles during HRTEM imaging. Sonication was limited to 1 min because extended sonication up to 30 min resulted in morphological changes that were observed in AFM, seen in Supplementary Figure 5. High-resolution images were taken with a JEOL 2100F field-emission TEM (FETEM) at 200 kV. Samples were imaged for brief time periods under low current density to prevent radiation-induced restructuring. Representative images of radiation-induced damage can be seen in Supplementary Figure 6.

**Liquid transmission electron microscopy**. All liquid TEM work was started within an hour after synthesis. Post-synthesized sample was diluted 10× in DI water approximately after synthesis and imaged using a Protochips Poseidon holder fitted with an E-chips liquid cell. The cell consists of two silicon chips, coated with a 50-nm-thick silicon nitride membrane, with windows of

approximately $500 \times 50 \ \mu m^2$. The spacer between the two chips, composed of 150-nm-thick SU-8, was deposited in a modified flow configuration.

A Tecnai F20 FETEM was used at 200 kV in bright-field mode at an electron dose under $500 \ e^- \ nm^{-2} \ s^{-1}$. Higher electron doses were used to induce bubble formation. Video was taken at 25 fps at $4k \times 4k$ resolution using a Gatan OneView camera. Sample was imaged under static conditions as well as in flow, under $200 \ \mu L \ h^{-1}$ flowrates of DI water. Particle sizes and positions were measured in imageJ.

**Atomic force microscopy.** The ethanol suspensions were sonicated for 1 min, dropped onto 1 mm quartz slides, and dried at room temperature. Height and phase maps were taken with a Bruker Dimension ICON AFM with a silicon SPM probe (MikroMasch NSC15) while tapping in air. Maps were flattened using a first- or second-order fit for each line scan and analyzed using Bruker Nanoscope Analysis.

The heights and diameters of at least 100 individual particles were measured for each sample to construct a reasonable particle size distribution. Particle diameter was approximated by treating the area of the particle as the area of a circle. Height was taken to be the difference of the maximum measured height and the surrounding height.

**X-ray diffraction.** The ethanol suspensions were deposited onto a non-reflective silicon wafer (100) and dried at ambient temperature. The powder XRD pattern was collected with a Brucker D8 Advance Powder Diffractometer using Cu K$\alpha$ radiation ($\lambda = 1.5418$ Å) at 40 kV from a $2\theta$ of 3–70° with a scanning resolution of $0.02° \ s^{-1}$. Crystallite size of the 110 peak was determined using the Scherrer equation[67].

**Peclet number calculation.** For a Brownian particle, Pe is given by Eq. 5, where $\eta_m$ is the medium's viscosity (Pas), $R_H$ is the hydrodynamic radius (m), $k_B$ is Boltzmann's constant ($m^2 \ kg \ s^{-2} \ K^{-1}$), $T$ is temperature (K), and $D_{B_0}$ is the Brownian self-diffusion coefficient ($m^2 \ s^{-1}$).

$$Pe = \frac{6\pi \eta_m R_H^3 \dot{\gamma}}{k_B T} = \frac{R_H^2 \dot{\gamma}}{D_{B0}} \tag{5}$$

The hydrodynamic radius of a disc $R_{H, \ disc}$ is given by Eqs. 6 and 7, where $H_{disc}$ is the disk thickness, $R_{disc}$ is the disc radius, and $\alpha$ is the aspect ratio (which is not to be confused with the definition of aspect ratio used in AFM particle analysis)[68].

$$\alpha = \frac{H_{disc}}{2 R_{disc}} \tag{6}$$

$$R_{H,disc} = \frac{3 R_{disc}}{2} \left[ \sqrt{1 + \alpha^2} + \frac{1}{\alpha} \ln \left( \alpha + \sqrt{1 + \alpha^2} \right) - \alpha \right]^{-1} \tag{7}$$

For discs 2 and 4 nm in radius with a thickness of 0.7 nm, Pe equals 0.07 and 0.46, respectively, at a shear rate of $3.16 \times 10^6 \ s^{-1}$. The rotational Peclet number is given by Eq. 8, where $D_r$ is the rotational diffusivity about the cylindrical axis[69].

$$Pe_r = \frac{\dot{\gamma}}{D_r} = \dot{\gamma} \left( \frac{3 k_B T}{32 \eta_m R_{disc}^3} \right)^{-1} \tag{8}$$

For disks 2 and 4 nm in radius, $Pe_r$ equals 0.06 and 0.47, respectively, at a shear rate of $3.16 \times 10^6 \ s^{-1}$.

**Characteristic growth time calculation.** The surface growth reaction is seen in Eq. 9, in which the addition of a cation $A$ to particle $B_i$ results in a larger particle, $B_{i+1}$, where $i$ represents the size. At high supersaturations, this is assumed to be irreversible.

$$A + B_i \rightarrow B_{i+1} \tag{9}$$

The rate of growth $r_{g, \ i+1}$ is the given by Eq. 10, where $K_i^g$ is the rate coefficient for a particle of size $i$ ($mol^{-1} \ m^3 \ s^{-1}$), $[A]$ is the concentration of cations ($mol \ m^{-3}$), and $[B_i]$ is the concentration of $B_i$ particles ($mol \ m^{-3}$). An alternate calculation of the growth rate is given in Supplementary Discussion - Calculation of Growth Rate using the Sherwood number.

$$r_{g,i+1} = K_i^g [A][B_i] \tag{10}$$

The characteristic growth timescale of a particle of the critical nucleus size $n$ ($R_{disc} = 2$ nm, $H_{disc} = 0.7$ nm) is then given as $\tau_{g,n}$, seen in Eq. 11, where $[A]_0$ and $[B_n]_0$ are the initial concentrations of cations and particles of the critical nucleus size, $B_n$. The initial concentration of cations is $[A]_0 = 0.75 \cdot 10^{-4} \ mol \ m^{-3}$ and the

initial concentration of nuclei is $[B_n]_0 = 2.0 \cdot 10^{-7} mol \ m^{-3}$, which is approximated from the average particle number concentration.

$$\tau_{g,n} = \left( K_n^g \sqrt{[A]_0 [B_n]_0} \right)^{-1} \tag{11}$$

At the micro-scale, the characteristic growth time is assumed to be diffusion-limited at high supersaturation. The rate coefficient is then determined by the rate of collision of the two species. This is given in Eq. 12, where $R_{H, \ A}$ and $R_{H,B_n}$ are the hydrodynamic radii of monomer $A$ and nucleus $B_n$ (m), $D_A$ and $D_{B_n}$ are their respective diffusivities ($m^2 \ s^{-1}$), and $N_A$ is Avogadro's constant ($mol^{-1}$).

$$K_n^g = 4\pi \left( R_{H,A} + R_{H,B_n} \right) \left( D_A + D_{B_n} \right) N_A \tag{12}$$

We assume that $R_{H,A} + R_{H,B_n} \cong R_{H,B_n} = 1.8 \cdot 10^{-9} m$ and $D_A = 1.020 \cdot 10^{-9} m^2 s^{-1}$[70]. $D_{B_n}$ is calculated from the Einstein relation in Eq. 13.

$$D_{B_n} = \frac{k_B T}{6\pi \eta_m R_{H,B_n}} = 1.364 \cdot 10^{-10} m^2 s^{-1} \tag{13}$$

Inserting the calculated diffusion coefficient into Eq. 12 yields $K_n^g = 1.55 \cdot 10^7 m^3 s^{-1} mol^{-1}$. Inserting $K_n^g$ and the known concentrations into Eq. 11 yields $\tau_g = 16.6 \cdot 10^{-3} s$.

**Characteristic aggregation time calculation.** For two Brownian particles, the binary collision rate coefficient $K_{ij}^A$ is given by Eq. 14, where $W_{ij}$, the stability coefficient for two particles of the same size, is equal to 1 at carbonate concentrations above 1 mmol[53], and $R_{H,i}$ and $R_{H,j}$ are the hydrodynamic diameters of the two particles of sizes $i$ and $j$, respectively.

$$K_{ij}^A = \frac{2 k_B T}{3 \eta_m W_{ij}} \left( R_{H,i} + R_{H,j} \right) \left( \frac{1}{R_{H,i}} + \frac{1}{R_{H,j}} \right) \tag{14}$$

The characteristic aggregation time for two particles of size $i$ in the absence of hydrodynamic forces is expressed by Eq. 15. For a binary collision between Brownian discs with a hydrodynamic radius of 2 nm, $\tau_{nn}^A = 0.68$ s.

$$\tau_{ii}^A = \left( K_{ii}^A [B_n]_0 N_A \right)^{-1} = \frac{3 \eta_m W_{11}}{8 k_B T [B_n]_0 N_A} \tag{15}$$

The scaling of the characteristic aggregation time for a binary encounter between two particles of the same size $i$ with a fixed potential ($\tau_{ii}^{A,Pe}$) is given by Eq. 16[49].

$$\tau_{ii}^{A,Pe} \sim \left( \frac{\eta_m \dot{\gamma} R_{H,i}^3}{k_B T} \right)^{-\frac{1}{2}} e^{\frac{-2\eta_m \dot{\gamma} R_H^3}{k_B T}} = \left( \frac{Pe}{6\pi} \right)^{-\frac{1}{2}} e^{-Pe/3\pi} \tag{16}$$

For a binary collision between discs of size $n$, we assume that as Pe$\rightarrow$0 the characteristic aggregation time is given by Eq. 15, such that $\tau_{nn}^{A,Pe} (\dot{\gamma} = 4.1 \cdot 10^4 s^{-1}, Pe = 9.5 \cdot 10^{-4}) = 0.68$ s. Therefore, $\tau_{nn}^{A,Pe} (\dot{\gamma} = 8.4 \cdot 10^5 s^{-1}, Pe = 1.9 \cdot 10^{-2}) = 0.15$ s and $\tau_{nn}^{A,Pe} (\dot{\gamma} = 3.2 \cdot 10^6 s^{-1}, Pe = 7.4 \cdot 10^{-2}) = 77$ ms.

## Data availability

Source data for plots and other findings are available from the corresponding author on request.

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

## Acknowledgements

We thank Dr. Kamran Yunus for his assistance in high-speed microscopic studies and Dr. Giorgio Divitini and Dr. Caterina Ducati for their assistance in liquid TEM studies. This project is funded by the National Research Foundation (NRF), Prime Minister's Office, Singapore under its Campus for Research Excellence and Technological Enterprise (CREATE) program as a part of the Cambridge Centre for Advanced Research and Education in Singapore Ltd (CARES). Funding for the liquid TEM holder was provided by the Engineering and Physical Sciences Research Council's Henry Royce Institute (EP/P024947/1).

## Author contributions

N.A.J. conceived, designed and performed the experiments, and wrote the manuscript. A.A.L. and H.C.Z. advised and contributed to the manuscript.

## Additional information

**Competing interests:** N.A.J. and A.A.L. filed a patent application for the reactor described (Singapore patent app. no. 10201801303T). H.C.Z. declares no competing interests.

