## [Peer Review File · Nature Communications]

Reviewers' comments:

Reviewer #1 (Remarks to the Author):

This interesting manuscript containing novel material and effective approach. I suggest publication of this manuscript after revision. My questions and remarks are listed below.

1. Lines 44,45: Authors write that “the OA of patchy trimer particles is accelerated under shear”. Of course for not too high values of shear.
2. Line 138: how diameter of not well-rounded particles is defined?
3. Line 140: AR is not defined by Eqn.5
4. Line 156: Suggestion that shear affects particle crystallinity: can be this due to increased mass transfer coefficient?
5. Lines 189: there can be effect of particle concentration on Brownian diffusivity.
6. Lines 191,192: in this context the attractive van der Waals forces can be mentioned.
7. Line 224: perhaps both mechanism; Ostwald ripening is well documented.
8. Line 330: divided by the average liquid velocity?

Page 20, Villermaux/Dushman reaction method: The kinetics of Villermaux/Dushman reaction system is applied. However, quantitative application of test reactions requires the kinetics to be known under the conditions of mixing. as shown more recently by Bourne (2008), Kölbl and Schmidt-Lehr (2010) and Kölbl et al., (2013), none of the available in the literature kinetic models of the Dushman reaction satisfies this requirement, which means that this reaction can only be used to rank different mixers or check effects of varying process conditions on mixing (when strictly the same concentrations solutions and the same flow rate ratio or volume ratio values are used in various systems) but should not be used for quantitative applications. These limitations should be pointed out in the manuscript. The simplest correction (but not complete) is to include effect of dissociation of sulphuric acid (see “Revision of the Dushman reaction kinetics for an improved micromixing characterization”. by P. Guichardon, N. Ibaseta, iscres.org/ISCRE24/abs/65_file_2_Page_Abstract) to make considerations more universal.

Bourne J.R., Comments on the iodide/iodate method for characterizing micromixing, *Chemical Engineering Journal*, **140**, 638-641, 2008.

Kölbl A. K., Desplantes V., Grundemann L., Scholl S., Kinetic investigation of the Dushman reaction at concentrations relevant to mixing studies in stirred tank reactors, *Chemical Engineering Science*, **93**, 47–54, 2013.

Kölbl A., Schmidt-Lehr S., The iodide iodate reaction method: the choice of the acid, *Chem. Eng. Sci.*, **65**, 1897–1901, 2010.

9. Line 424, Eqn (2): should be R_h instead of a . It would be good to extend Eqn (2)_adding = $\frac{R_h^2 \dot{\gamma}}{D_{B0}}$
10. Eqs (4) to (7): Why for diffusion limited growth the Smoluchowski equation for perikinetic aggregation is used and description of mass transfer based on the concept mass transfer coefficient and the Sherwood number is not used?

11. Line 441, Eqn (7) for A and B_i concentrations in molm⁻³ there should be N_A^2 instead of N_A to have product of number concentrations when combining equations (5) and (7) for perikinetic aggregation.

Jerzy Baldyga

Reviewer #2 (Remarks to the Author):

This manuscript reports on a very interesting study of the growth and shear-induced aggregation of disk-shaped LDH particles in a microreactor device. Controlling the growth of anisotropic nanocrystals is a highly topical issue and this study characterises the effect of shear on this for an interesting and relevant system. The use of AFM and XRD, alongside of liquid TEM, provides a clear picture of how shear affects the particle and aggregate size, with intermediate shear rates leading to monodisperse particles that aggregate via oriented attachment. The observation of particle aggregation and oriented attachment under flow using liquid TEM does break new ground I believe, although I agree with the authors that the effects of local e-beam heating are an important caveat to these observations.

However, there appear to be an unusually large number of mistakes (detailed below) in their descriptions of the equations used and this detracts substantially from their presentation. My recommendation is that the paper be revised substantially to improve the presentation of their model - it is sufficiently sloppy that it is possible they have obscured mistakes in their computations. I think this paper could be important to this field, but this aspect of the work needs considerable improvement.

I detail my comments and queries for the authors below.

Pg 9, Line 140, the authors refer to Eqn 5 with regards to the aspect ratios measured by AFM. This seems to be a mistake or at least is far too brief. Eqn 5 concerns the particle growth rate, which may be related to the aspect ratio, but needs more explanation.

On Pg 16, the discuss the possibility that the 2.14 nm-sized particles represent the size of the critical nuclei, above which particles would grow by surface addition of aggregation. I think needs some more discussion as the particle aggregates don't seem to anneal substantially once attachment occurs. This may suggest some sort of surface stabilisation that is more complex than simply reaching the size of the critical nuclei. I note that annealing is discussed in the supplementary material (Fig 4 shows this occurring under the e-beam) - perhaps this should be discussed further there.

Pg 21, Eqn 1 - this equation describes the engulfment time and is very poorly described in the text. The engulfment time is the square root of the ratio dynamic viscosity to the rate of turbulent energy dissipation. I also suspect the coefficient is wrong - I believe it should be 12 not 17. It is not entirely clear whether they need this equation, but if they do then they need to justify its use by computing the Schmidt number.

Pg 24, Eqn 2 - The authors introduce the hydrodynamic radius R_H but then define the Peclet number in terms of a , when presumably they mean R_H .

Pg 25, Eqn 3 - which axis is the rotation about?

Pg 25, Eqn 6 - what do the subscript zeroes mean on the concentrations? It later becomes apparent that these are initial concentrations, but it should be made clear when they are introduced. What does it mean when they switch to subscript n for species B in this equation?

From Eqn 2 and on, the authors switch between describing the particles as discs and as spheres. I think it would be helpful if they could be clearer as to why in each instance. I found this very confusing.

After Eqn 8, they say “Equation eight yields ...” but what they mean is the diffusion constant computed in Eqn 8, when inserted into Eqn 7 yields a rate constant ... but then they switch back to the subscript i for the rate constant!

Reviewer #3 (Remarks to the Author):

The manuscript deals with a subject still open in the literature: how to control the assembly of lamellar nanomaterials. The authors have developed a very interesting method in which they control the flow of the solution in order to allow the alignment of the nanoparticles. The idea is very interesting and undoubtedly shocking. However, the results presented are not shocking enough.

Here are some of the shortcomings noted:

a) The images of HRTEM are of low quality (Figure 3). Apparently, it has a strong delocalization and defocuses. Are these samples sensitive to the electron beam? The authors should present better images;

b) The kinetic analysis assumes the rigid sphere model. However, the particles are lamellar and 2D (like a disc). This should cause a considerable error in the values of P_e , P_r , and characteristic growth time. Thus the analysis can be wronged. It would be interesting to adopt a more appropriate geometric factor to the model (such as a disc);

c) Finally, the image sequence of Figure 4 is fabulous. However, the authors did not explain the process of attracting and aligning the particles. This explanation is fundamental to understanding the OA process. The authors also did not provide an analysis of the surface charges (by potential zeta analysis for instance). This analysis is fundamental to understand the process of alignment of the nanoparticles.

In the present format, this manuscript is not suitable to be published

We are most grateful for excellent constructive comments and suggestions. These were most instructive in improving the quality of the manuscript. We have addressed all comments and are re-submitting the new version of the manuscript. Below please find responses to all comments.

Reviewer #1

This interesting manuscript containing novel material and effective approach. I suggest publication of this manuscript after revision. My questions and remarks are listed below.

1. Lines 44,45: Authors write that "the OA of patchy trimer particles is accelerated under shear". Of course for not too high values of shear.

Agreed, see correction at Line 45.

2. Line 138: how diameter of not well-rounded particles is defined?

See clarification at Line 138.

3. Line 140: AR is not defined by Eqn.5

See clarification at Line 140.

4. Line 156: Suggestion that shear affects particle crystallinity: can be this due to increased mass transfer coefficient?

We have presented our explanation for the effect on particle crystallinity in Section 5. Shear does increase mass transfer, but more than this it plays a role in oriented attachment, which is not described by bulk mass transfer coefficients.

5. Lines 189: there can be effect of particle concentration on Brownian diffusivity.

See clarification at Line 205.

6. Lines 191,192: in this context the attractive van der Waals forces can be mentioned.

Agreed, see mention at line 208.

7. Line 224: perhaps both mechanism; Ostwald ripening is well documented.

Agreed, see correction at line 242.

8. Line 330: divided by the average liquid velocity?

Agreed, see correction at line 405.

Page 20, Villermaux/Dushman reaction method: The kinetics of Villermaux/Dushman reaction system is applied. However, quantitative application of test reactions requires the kinetics to be known under the conditions of mixing. as shown more recently by Bourne (2008), Kölbl and Schmidt-Lehr (2010) and Kölbl et al., (2013), none of the available in the literature kinetic models of the Dushman reaction satisfies this requirement, which means that this reaction can only be used to rank different mixers or check effects of varying process conditions on mixing (when strictly the same concentrations solutions and the same flow rate ratio or volume ratio values are used in various systems) but should not be used for quantitative applications. These limitations should be pointed out in the manuscript. The simplest correction (but not complete) is to include effect of dissociation of sulphuric acid (see "Revision of the Dushman reaction kinetics for an improved micromixing characterization". by P. Guichardon, N. Ibaseta, iscre.org/ISCRE24/abs/65_file_2_Page_Abstract) to make considerations more universal.

Bourne J.R., Comments on the iodide/iodate method for characterizing micromixing, *Chemical Engineering Journal*, 140, 638-641, 2008.

Kölbl A. K., Desplantes V., Grundemann L., Scholl S., Kinetic investigation of the Dushman reaction at concentrations relevant to mixing studies in stirred tank reactors, *Chemical Engineering Science*, 93, 47-54, 2013.

Kölbl A., Schmidt-Lehr S., The iodide iodate reaction method: the choice of the acid, *Chem. Eng. Sci.*, 65, 1897-1901, 2010.

Agreed. This is the reason that we also measured mixing via fluorescence microscopy. By comparing the two we have provided a more universal estimation of mixing time. This analysis has been added to the revision.

9. Line 424, Eqn (2): should be R_H instead of a . It would be good to extend Eqn (2) adding $= \frac{R_H^2 \bar{v}}{D_{Bn}}$

Agreed, see correction at Equation 2.

10. Eqs (4) to (7): Why for diffusion limited growth the Smoluchowski equation for perikinetic aggregation is used and description of mass transfer based on the concept mass transfer coefficient and the Sherwood number is not used?

We have used the Smoluchowski equation to directly provide a growth rate in terms of mol/s from a first principles approach, which is also similar to the approach used to calculate growth rates via aggregation under shear. If we calculate the growth rate using the semi-empirical approach using the Sherwood number we still obtain a similar result for the growth rate. See calculations below.

To use the Sherwood approach we consider a hexagonal particle with side-length ($a = 2$ nm), which we use as the characteristic length scale. The Sherwood number is expressed with the correlation in Equation 1 (Frossling, 1938), where k_d is the mass transfer coefficient, D_A is the diffusion coefficient of solute species A ($1.02 \cdot 10^{-9} \text{ m}^2 \text{ s}^{-1}$), $Re = u_s a \rho / \mu$ is Reynolds number (u_s is the particle slip velocity, ρ is the solution density, μ is the solution dynamic viscosity) and $Sc = \mu / \rho D_A$ is the Schmidt number. For very small particles (< 10 nm) the slip velocity of the particle (u_s), such that $Sh \approx 2$.

$$Sh = \frac{k_d a}{D_A} = 2 + 1.10 Re^{\frac{1}{2}} Sc^{\frac{1}{3}} \approx 2 \quad (1)$$

We can then use the semiempirical relation in Equation 2 to obtain G , which is the growth rate of the particle along its characteristic growth dimension in m s^{-1} , where ρ_c is the crystal density ($2.05 \cdot 10^3 \text{ mol m}^{-3}$) and we assume the difference between the initial bulk concentration of A ($[A]_0 = 0.75 \cdot 10^{-3} \text{ mol m}^{-3}$) is much greater than the crystal-surface interface concentration of A ($[A]_i$).

$$G = \frac{2k_d}{\rho_c} ([A]_0 - [A]_i) \approx \frac{4D_A}{\rho_c a} [A]_0 \quad (2)$$

To then convert G into $r_{g,n}$, which is in mol s^{-1} , we use Equation 3, which converts the growth rate of a particle of size n along a into the mass growth rate of hexagonal particles, where H is the hexagon thickness (0.7 nm), N_A is Avogadro's number ($6.02 \cdot 10^{23} \text{ mol}^{-1}$) and $[B_n]$ is the concentration of particles of size n ($2 \cdot 10^{-7} \text{ mol m}^{-3}$)

$$r_{g,n} = 3\sqrt{3} H a G \rho_c N_A [B] \quad (3)$$

Inserting $r_{g,n}$ into equation 4, we solve for the kinetic growth rate constant, which is very similar to the previously calculated $1.42 \cdot 10^7 \text{ m}^3 \text{ s}^{-1} \text{ mol}^{-1}$, considering the approximations involved.

$$K_n^g = \frac{r_{g,n}}{[A][B_n]} = 9.0 \cdot 10^6 \text{ m}^3 \text{ s}^{-1} \text{ mol}^{-1} \quad (4)$$

Frossling, N. (1938). *Gerlands Beitr. Geophys.* **52**, 170

11. Line 441, Eqn (7) for A and B_i concentrations in mol m^{-3} there should be N_A^2 instead of N_A to have product of number concentrations when combining equations (5) and (7) for perikinetic aggregation.

Our units are consistent with the calculations provided. See unit analysis below:

Equation 9 (previously 7):

$$K_n^g = 4\pi(R_{H,A} + R_{H,B_n})(D_A + D_{B_n})N_A [=](m) \cdot (m^2 s^{-1}) \cdot (mol^{-1}) = m^3 s^{-1} mol^{-1}$$

Equation 7 (previously 5):

$$r_{g,i+1} = K_i^g [A][B_i] [=](m^3 s^{-1} mol^{-1}) \cdot (mol \cdot m^{-3}) \cdot (mol \cdot m^{-3}) = mol \cdot m^{-3} s^{-1}$$

Equation 8 (previously 6):

$$\tau_g = (K_n^g \sqrt{[A]_0 [B_n]_0})^{-1} [=] \left(mol \cdot m^{-3} s^{-1} \cdot \sqrt{(mol \cdot m^{-3}) \cdot (mol \cdot m^{-3})} \right)^{-1} = s$$

Reviewer #2

This manuscript reports on a very interesting study of the growth and shear-induced aggregation of disk-shaped LDH particles in a microreactor device. Controlling the growth of anisotropic nanocrystals is a highly

topical issue and this study characterises the effect of shear on this for an interesting and relevant system. The use of AFM and XRD, alongside of liquid TEM, provides a clear picture of how shear affects the particle and aggregate size, with intermediate shear rates leading to monodisperse particles that aggregate via oriented attachment. The observation of particle aggregation and oriented attachment under flow using liquid TEM does break new ground I believe, although I agree with the authors that the effects of local e-beam heating are an important caveat to these observations.

However, there appear to be an unusually large number of mistakes (detailed below) in their descriptions of the equations used and this detracts substantially from their presentation. My recommendation is that the paper be revised substantially to improve the presentation of their model - it is sufficiently sloppy that it is possible they have obscured mistakes in their computations. I think this paper could be important to this field, but this aspect of the work needs considerable improvement.

I detail my comments and queries for the authors below.

1. Pg 9, Line 140, the authors refer to Eqn 5 with regards to the aspect ratios measured by AFM. This seems to be a mistake or at least is far too brief. Eqn 5 concerns the particle growth rate, which may be related to the aspect ratio, but needs more explanation.

This was a mistake. The formula for aspect ratio is included in the revised text

2. On Pg 16, they discuss the possibility that the 2.14 nm-sized particles represent the size of the critical nuclei, above which particles would grow by surface addition or aggregation. I think needs some more discussion as the particle aggregates don't seem to anneal substantially once attachment occurs. This may suggest some sort of surface stabilisation that is more complex than simply reaching the size of the critical nuclei. I note that annealing is discussed in the supplementary material (Fig 4 shows this occurring under the e-beam) - perhaps this should be discussed further there.

Agreed. Solvent-solute-surface interactions probably play a large part in stabilizing this particle size. We have included additional discussion of this in the revised manuscript.

3. Pg 21, Eqn 1 - this equation describes the engulfment time and is very poorly described in the text. The engulfment time is the square root of the ratio dynamic viscosity to the rate of turbulent energy dissipation. I also suspect the coefficient is wrong - I believe it should be 12 not 17. It is not entirely clear whether they need this equation, but if they do then they need to justify its use by computing the Schmidt number.

The coefficient is 17 for the characteristic engulfment time (Baldyga & Bourne, 1984; Baldyga, 2016). The reviewer may be referring to the coefficient for the hydrodynamic lifetime of an eddy (12), which is a similar timescale, but represents a different quantity. A better description for the engulfment time is added in the revision. We have included the value of the Schmidt number to justify the use of this equation.

Baldyga J, Bourne JR. Simplification of Micromixing Calculations .2. New Applications. *Chem Eng J Bioch Eng* 1989, **42**(2): 93-101.

Baldyga J. Mixing and Fluid Dynamics Effects in Particle Precipitation Processes. *Kona Powder Part J* 2016(33): 127-149.

4. Pg 24, Eqn 2 - The authors introduce the hydrodynamic radius R_H but then define the Peclet number in terms of a , when presumably they mean R_H .

Agreed, see correction at Eqn 2.

5. Pg 25, Eqn 3 - which axis is the rotation about?

The cylindrical axis. See clarification in the revision.

6. Pg 25, Eqn 6 - what do the subscript zeroes mean on the concentrations? It later becomes apparent that these are initial concentrations, but it should be made clear when they are introduced.

Correct, this is initial concentrations. See clarification in the revision.

7. What does it mean when they switch to subscript n for species B in this equation?

The n subscript refers to the size of a nucleus or primary particle, such that B_n represents a B particle of size n . This has been clarified in the revision.

8. From Eqn 2 and on, the authors switch between describing the particles as discs and as spheres. I think it would be helpful if they could be clearer as to why in each instance. I found this very confusing.

We have clarified in the text that the use of terms

9. After Eqn 8, they say "Equation eight yields ..." but what they mean is the diffusion constant computed in Eqn 8, when inserted into Eqn 7 yields a rate constant ... but then they switch back to the subscript i for the rate constant!

This has been modified.

Reviewer #3 (Remarks to the Author):

The manuscript deals with a subject still open in the literature: how to control the assembly of lamellar nanomaterials. The authors have developed a very interesting method in which they control the flow of the solution in order to allow the alignment of the nanoparticles. The idea is very interesting and undoubtedly shocking. However, the results presented are not shocking enough.

Here are some of the shortcomings noted:

a) The images of HRTEM are of low quality (Figure 3). Apparently, it has a strong delocalization and defocuses. Are these samples sensitive to the electron beam? The authors should present better images.

LDH nanoplatelets are very sensitive to electron beam irradiation. These effects have been documented in the manuscript and are included in the supporting information section in Supplementary Figure 6. The HRTEM images are not aberration corrected, so atomic level resolution could not be achieved. However, resolution is high-enough to provide information about the crystalline lattice and domain size, which we use to support our claims.

b) The kinetic analysis assumes the rigid sphere model. However, the particles are lamellar and 2D (like a disc). This should cause a considerable error in the values of Pe , Per , and characteristic growth time. Thus the analysis can be wronged. It would be interesting to adopt a more appropriate geometric factor to the model (such as a disc);

Agreed, the analysis has been adjusted to assume disc-shaped particles.

c) Finally, the image sequence of Figure 4 is fabulous. However, the authors did not explain the process of attracting and aligning the particles. This explanation is fundamental to understanding the OA process. The authors also did not provide an analysis of the surface charges (by potential zeta analysis for instance). This analysis is fundamental to understand the process of alignment of the nanoparticles.

Agreed, further analysis has been added in the revision.

Additional Changes:

In reviewing the calculations, we corrected calculation of aggregation time. We corrected these calculations as below:

Equation 12

Coefficient at the denominator of equation 12 should be 8

Equation 13

Numerators in equation should not contain π . The expression $\left(\frac{\eta_m \dot{\gamma} R_H^3}{k_B T}\right)^{-\frac{1}{2}} e^{-\frac{2\eta_m \dot{\gamma} R_H^3}{k_B T}}$ is a scaling factor instead of a direct expression for the aggregation time. We conduct the new analysis as was done in the original paper that introduced this scaling factor (Zaccone, 2009). In the revision we clarify that the aggregation model is only an estimate, as orientation effects are not included.

Reviewers' comments:

Reviewer #1 (Remarks to the Author):

In my opinion this manuscript can be now published as it is or after minor revision; I leave this to authors discretion.

Regarding authors answers to my questions and remarks from my first review:

1 to 9: OK.

10. Eqs (4) to (7): Why for diffusion limited growth the Smoluchowski equation for perikinetic aggregation is used and description of mass transfer based on the concept mass transfer coefficient and the Sherwood number is not used?

Rebuttal:

We have used the Smoluchowski equation to directly provide a growth rate in terms of mol/s from a first principles approach, which is also similar to the approach used to calculate growth rates via aggregation under shear. If we calculate the growth rate using the semi-empirical approach using the Sherwood number we still obtain a similar result for the growth rate. See calculations below.

To use the Sherwood approach we consider a hexagonal particle with side-length ($a = 2$ nm), which we use as the characteristic length scale. The Sherwood number is expressed with the correlation in Equation 1 (Frossling, 1938), where k_d is the mass transfer coefficient, D_A is the diffusion coefficient of solute species A ($1.02 \cdot 10^{-9} \text{ m}^2 \text{ s}^{-1}$), $Re = u_s a \rho / \mu$ is Reynolds number (u_s is the particle slip velocity, ρ is the solution density, μ is the solution dynamic viscosity) and $Sc = \mu / \rho D_A$ is the Schmidt number. For very small particles (< 10 nm) the slip velocity of the particle (u_s), such that $Sh \approx 2$.

$$Sh = \frac{k_d a}{D_A} = 2 + 1.10 Re^{\frac{1}{2}} Sc^{\frac{1}{3}} \approx 2 \quad (1)$$

We can then use the semiempirical relation in Equation 2 to obtain G , which is the growth rate of the particle along its characteristic growth dimension in m s^{-1} , where ρ_c is the crystal density ($2.05 \cdot 10^3 \text{ mol m}^{-3}$) and we assume the difference between the initial bulk concentration of A ($[A]_0 = 0.75 \cdot 10^{-3} \text{ mol m}^{-3}$) is much greater than the crystal-surface interface concentration of A ($[A]_i$).

$$G = \frac{2 k_d}{\rho_c} ([A]_0 - [A]_i) \approx \frac{4 D_A}{\rho_c a} [A]_0 \quad (2)$$

To then convert G into $r_{g,n}$, which is in mol s^{-1} , we use Equation 3, which converts the growth rate of a particle of size n along a into the mass growth rate of hexagonal particles, where H is the hexagon thickness (0.7 nm), N_a is Avogadro's number ($6.02 \cdot 10^{23} \text{ mol}^{-1}$) and $[B_n]$ is the concentration of particles of size n ($2 \cdot 10^{-7} \text{ mol m}^{-3}$)

$$r_{g,n} = 3\sqrt{3} H a G \rho_c N_a [B] \quad (3)$$

Inserting $r_{g,n}$ into equation 4, we solve for the kinetic growth rate constant, which is very similar to the previously calculated $1.42 \cdot 10^7 \text{ m}^3 \text{ s}^{-1} \text{ mol}^{-1}$, considering the approximations involved.

$$K_n^g = \frac{r_{g,n}}{[A][B_n]} = 9.0 \cdot 10^6 \text{ m}^3 \text{ s}^{-1} \text{ mol}^{-1} \quad (4)$$

Frossling, N. (1938). *Gerlands Beitr. Geophys.* 52, 170

I have here 3 comments:

A. Definition of Sh in equation (1) above: should be based on sphere diameter not radius to have 2, $Sh = \frac{k_d d}{D_A} = \frac{k_d 2a}{D_A} = 2$, hence estimation of the time constant, equation (4), should be corrected.

B. eq.(9) in the new manuscript is limited to spherical particles and neglects among other limitations the Gibbs-Thomson effect, important for development of particle shape.

C. Presented approach based on eq.(9) can be used for time scale estimation but not for growth rate predictions; the choice of the time scale estimation method being left to authors.

11. Line 441, Eqn (7) for A and B_i concentrations in molm⁻³ there should be N_i instead of N_A to have product of number concentrations when combining equations (5) and (7) for perikinetic aggregation

Our units are consistent with the calculations provided. See unit analysis below:

Equation 9 (previously 7):

$$K_n^g = 4\pi(R_{H,A} + R_{H,B_n})(D_A + D_{B_n})N_A [=](m) \cdot (m^2s^{-1}) \cdot (mol^{-1}) = m^3s^{-1}mol^{-1}$$

Equation 7 (previously 5):

$$\tau_{g,i+1} = K_i^g[A][B_i] [=](m^3s^{-1}mol^{-1}) \cdot (mol \cdot m^{-3}) \cdot (mol \cdot m^{-3}) = mol \cdot m^{-3}s^{-1}$$

Equation 8 (previously 6):

$$\tau_g = (K_n^g \sqrt{[A]_0[B_n]_0})^{-1} [=] (mol \cdot m^{-3}s^{-1} \cdot \sqrt{(mol \cdot m^{-3}) \cdot (mol \cdot m^{-3})})^{-1} = s$$

Agree. I was thinking about original equation by Smoluchowski, where both concentrations are expressed as particle number concentrations.

Additional remark:

The shear rate based on the gradient of velocity (estimated using “the film thickness and velocity or gas and liquid flow rates), is used in this manuscript both in discussions and definitions. For the purpose of problems considered in this work it would be good to decompose the gradient tensor into the rate-of-strain or deformation rate tensor and the rate-of-rotation tensor and discuss consequences of results for mechanism of aggregation.

Reviewer #2 (Remarks to the Author):

The authors have addressed my concerns with their previous manuscript with this revision. Most of my concerns were related to the presentation of their model, and I see that this has been substantially improved. Also, in my opinion, they have successfully addressed the concerns of the other referees with respect to the model. As I said in my first review they report on a very interesting study of the growth and shear-induced aggregation of disk-shaped LDH particles in a microreactor device. Controlling the growth of anisotropic nanocrystals is a highly topical issue and this study characterises the effect of shear on this for an interesting and relevant system. The use of AFM and XRD, alongside of liquid TEM, provides a clear picture of how shear affects the particle and aggregate size, with intermediate shear rates leading to monodisperse particles that aggregate via oriented attachment. The observation of particle aggregation and oriented attachment under flow using liquid TEM breaks new ground. My recommendation is that the revised manuscript should be published.

Reviewer #3 (Remarks to the Author):

In this new manuscript version, the authors worked well and provide a modified version where the points with deficiency were corrected. In this version, the manuscript is suitable for publication. However, I insist the authors can provide a better HRTEM image (Fig 3 in the manuscript), with a lower electron delocalization level. Even for a sample with a low tolerance to electron irradiation damage, the authors can obtain such an image. Finally, just to remind the authors, an HRTEM microscope also possesses atomic resolution. A TEM with Cs correction will improve the image resolution (to sub-angstrom resolution levels) and decrease the delocalization.

We are most grateful to the positive comments on our revisions and to the further suggestions of the minor revisions. We have followed all the suggestions and persevered with obtaining good HRTEM images, as was strongly suggested by one of the referees. We were able to obtain good quality HRTEM images and would hope that in the current state the paper is acceptable.

Reviewer #1:

In my opinion this manuscript can be now published as it is or after minor revision; I leave this to authors discretion. Regarding authors answers to my questions and remarks from my first review:

1. 1 to 9: OK.
2. 10. Eqs (4) to (7): Why for diffusion limited growth the Smoluchowski equation for perikinetic aggregation is used and description of mass transfer based on the concept mass transfer coefficient and the Sherwood number is not used?

I have here 3 comments [in response to rebuttal]:

i. Definition of Sh in equation (1) above: should be based on sphere diameter not radius to have 2, $Sh = \frac{k_a d}{D_A} = \frac{k_a 2a}{D_A} = 2$, hence estimation of the time constant, equation (4), should be corrected.

This correction has been made. This calculation will be made available in the supplementary section. Reference has been made in the manuscript at line 308.

We have used the Smoluchowski equation to directly provide a growth rate in terms of mol/s from a first principles approach, which is also similar to the approach used to calculate growth rates via aggregation under shear. If we calculate the growth rate using the semi-empirical approach using the Sherwood number we still obtain a similar result for the growth rate. See calculations below.

To use the Sherwood approach we consider a hexagonal particle with side-length ($a = 2$ nm), which we use as the characteristic length scale. The Sherwood number is expressed with the correlation in Equation 1 (Frossling, 1938), where k_a is the mass transfer coefficient, D_A is the diffusion coefficient of solute species A ($1.02 \cdot 10^{-9} \text{ m}^2 \text{ s}^{-1}$), $Re = u_s a \rho / \mu$ is Reynolds number (u_s is the particle slip velocity, ρ is the solution density, μ is the solution dynamic viscosity) and $Sc = \mu / \rho D_A$ is the Schmidt number. For very small particles (< 10 nm) the slip velocity of the particle (u_s), such that $Sh \approx 2$.

$$Sh = \frac{k_a 2a}{D_A} = 2 + 1.10 Re^{\frac{1}{2}} Sc^{\frac{1}{3}} \approx 2 \quad (1)$$

We can then use the semiempirical relation in Equation 2 to obtain G , which is the growth rate of the particle along its characteristic growth dimension in m s^{-1} , where ρ_c is the crystal density ($2.05 \cdot 10^3 \text{ mol m}^{-3}$) and we assume the difference between the initial bulk concentration of A ($[A]_0 = 0.75 \cdot 10^{-3} \text{ mol m}^{-3}$) is much greater than the crystal-surface interface concentration of A ($[A]_i$).

$$G = \frac{2k_a}{\rho_c} ([A]_0 - [A]_i) \approx \frac{2D_A}{\rho_c a} [A]_0 \quad (2)$$

To then convert G into $r_{g,n}$, which is in mol s^{-1} , we use Equation 3, which converts the growth rate of a particle of size n along a into the mass growth rate of hexagonal particles, where H is the hexagon

thickness (0.7 nm), N_a is Avogadro's number ($6.02 \cdot 10^{23} \text{ mol}^{-1}$) and $[B_n]$ is the concentration of particles of size n ($2 \cdot 10^{-7} \text{ mol m}^{-3}$).

$$r_{g,n} = 3\sqrt{3}HaG\rho_c N_a [B] \quad (3)$$

Inserting $r_{g,n}$ into equation 4, we solve for the kinetic growth rate constant, which is very similar to the previously calculated $1.42 \cdot 10^7 \text{ m}^3 \text{ s}^{-1} \text{ mol}^{-1}$, considering the approximations involved.

$$K_n^g = \frac{r_{g,n}}{[A][B_n]} = 4.5 \cdot 10^6 \text{ m}^3 \text{ s}^{-1} \text{ mol}^{-1} \quad (4)$$

Frossling, N. (1938). *Gerlands Beitr. Geophys.* **52**, 170

- ii. **B. eq.(9) in the new manuscript is limited to spherical particles and neglects among other limitations the Gibbs-Thomson effect, important for development of particle shape.**
Agreed. Equation 9 in the new manuscript is used to just to calculate initial particle growth time scale and can be applied to non-spherical particles in this case by using hydrodynamic diameter. It is not being used to quantitatively determine particle shape evolution, which is outside the scope of this study.

- iii. **Presented approach based on eq.(9) can be used for time scale estimation but not for growth rate predictions; the choice of the time scale estimation method being left to authors.**

This is correct. As stated in the previous comment, we are using this equation simply to conduct time scale estimation, which is used in our analysis. A more detailed model is necessary for quantitative growth rate predictions, involving the effect of hydrodynamic shear rate on oriented attachment/aggregation of highly anisotropic structures.

3. 11. Line 441, Eqn (7) for A and B_i concentrations in molm^{-3} there should be N_A^2 instead of N_A to have product of number concentrations when combining equations (5) and (7) for perikinetic aggregation.

Agree [in response to rebuttal]. I was thinking about original equation by Smoluchowski, where both concentrations are expressed as particle number concentrations.

4. Additional remark:

The shear rate based on the gradient of velocity (estimated using “the film thickness and velocity or gas and liquid flow rates), is used in this manuscript both in discussions and definitions. For the purpose of problems considered in this work it would be good to decompose the gradient tensor into the rate-of-strain or deformation rate tensor and the rate-of-rotation tensor and discuss consequences of results for mechanism of aggregation.

The velocity field in this case is similar to simple shear flow. The analytical solution for the annular flow velocity distribution in two-dimensional coordinates, assuming that flow is axisymmetric, smooth and laminar, is given in equations 1 and 2, where u_1 and u_2 are the velocities in the axial and radial directions (m/s), x_1 is the axial distance, x_2 is the radial distance

from the tube axis (m), μ is the liquid viscosity (Pa·s) and B is the axial pressure gradient (Pa m⁻¹)¹.

$$u_1 = -\frac{BR^2}{4\mu} \left[1 - \left(\frac{x_2}{R} \right)^2 \right] \quad (1)$$

$$u_2 = 0 \quad (2)$$

Decomposing the gradient tensor $\frac{\partial u_i}{\partial x_j}$ into the rate-of-strain tensor \mathbf{S} and rate-of-rotation tensor \mathbf{R} with equations 3 - 5 yields the tensors in Equation 6 and 7.

$$\frac{\partial u_i}{\partial x_j} = \mathbf{S}_{ij} + \mathbf{R}_{ij} \quad (3)$$

$$\mathbf{S}_{ij} = \frac{1}{2} \left(\frac{\partial u_i}{\partial x_j} + \frac{\partial u_j}{\partial x_i} \right) \quad (4)$$

$$\mathbf{R}_{ij} = \frac{1}{2} \left(\frac{\partial u_i}{\partial x_j} - \frac{\partial u_j}{\partial x_i} \right) \quad (5)$$

$$\mathbf{S} = \begin{pmatrix} 0 & \frac{BR^2}{2\mu} x_2 \\ \frac{BR^2}{2\mu} x_2 & 0 \end{pmatrix} \quad (6)$$

$$\mathbf{R} = \begin{pmatrix} 0 & \frac{BR^2}{2\mu} x_2 \\ -\frac{BR^2}{2\mu} x_2 & 0 \end{pmatrix} \quad (7)$$

For a thin liquid layer (a small range of R) the rate of strain and rotational component are relatively constant over the volume of the liquid, compared to other flow regimes like pipe flow or Taylor flow. For this reason, the effects on aggregation are similar to that of simple shear flow, which are well studied for hard spheres and are currently areas of investigation for soft matter and complex-shaped particles.

For anisotropic particles, at shear rates at the order of $Pe_r \sim 1$ the rotational component will help control the orientation of anisotropic particles in flow. The orbits of disc-like particles about their axis will slow and become periodic, with becoming slowest when oriented close to the flow gradient direction². The pair-interaction distribution will be inhomogeneous, such that interactions along compressive axis are favoured and convected away along the extensional axis (Foss, 1999). By orienting particles relative to one another oriented aggregation is more favorable.

While shear may accelerate aggregation, above a certain threshold, stresses on aggregates will be high enough to cause breakup. Shear stress exerted by simple shear flow will cause breakup along the extensional axis and aggregate along the compressive axis, which may cause aggregate anisotropy³. This would further enhance the anisotropy of disc aggregates, due to their orientation in flow. This is why high shear rates used in this study resulted in thin aggregates.

At certain flowrates Kelvin-Helmholtz instabilities will occur, introducing deviations from the smooth, laminar analytical solution. Turbulence created by instabilities will cause deviations in particle velocities and trajectories from their behavior in simple shear. For this reason numerical simulations and experimental observation are more reliable in determining flow characteristics, and predicting the occurrence of instabilities.

- 1 Guo, Z., Fletcher, D. F. & Haynes, B. S. Numerical simulation of annular flow hydrodynamics in microchannels. *Computers & Fluids* **133**, 90-102, doi:<https://doi.org/10.1016/j.compfluid.2016.04.017> (2016).
- 2 Mewis, J. & Wagner, N. J. *Colloidal Suspension Rheology*. (Cambridge University Press, 2012).
- 3 Hoekstra, H., Vermant, J., Mewis, J. & Fuller, G. G. Flow-Induced Anisotropy and Reversible Aggregation in Two-Dimensional Suspensions. *Langmuir* **19**, 9134-9141, doi:10.1021/la034582k (2003).

We have added this to the manuscript/supplementary section. See reference to supplementary information at line 308.

Reviewer #2 (Remarks to the Author):

The authors have addressed my concerns with their previous manuscript with this revision. Most of my concerns were related to the presentation of their model, and I see that this has been substantially improved. Also, in my opinion, they have successfully addressed the concerns of the other referees with respect to the model. As I said in my first review they report on a very interesting study of the growth and shear-induced aggregation of disk-shaped LDH particles in a microreactor device. Controlling the growth of anisotropic nanocrystals is a highly topical issue and this study characterises the effect of shear on this for an interesting and relevant system. The use of AFM and XRD, alongside of liquid TEM, provides a clear picture of how shear affects the particle and aggregate size, with intermediate shear rates leading to monodisperse particles that aggregate via oriented attachment. The observation of particle aggregation and oriented attachment under flow using liquid TEM breaks new ground. My recommendation is that the revised manuscript should be published.

Reviewer #3 (Remarks to the Author):

In this new manuscript version, the authors worked well and provide a modified version where the points with deficiency were corrected. In this version, the manuscript is suitable for publication. However, I insist the authors can provide a better HRTEM image (Fig 3 in the manuscript), with a lower electron delocalization level. Even for a sample with a low tolerance to electron irradiation damage, the authors can obtain such an image.

Finally, just to remind the authors, an HRTEM microscope also possesses atomic resolution. A TEM with Cs correction will improve the image resolution (to sub-angstrom resolution levels) and decrease the delocalization.

We have included higher quality HRTEM images in Figure 3. We found that by using SiO coated TEM substrates, LDH particles are more stable under higher electron doses.